# Reducing Information Bottleneck for Weakly Supervised Semantic Segmentation

**Jungbeom Lee**[1]        **Jooyoung Choi**[1]        **Jisoo Mok**[1]        **Sungroh Yoon**[1,2,*]

[1] Department of Electrical and Computer Engineering, Seoul National University
[2] Interdisciplinary Program in Artificial Intelligence, Seoul National University
{jbeom.lee93, jy_choi, magicshop1118, sryoon}@snu.ac.kr

## Abstract

Weakly supervised semantic segmentation produces pixel-level localization from class labels; however, a classifier trained on such labels is likely to focus on a small discriminative region of the target object. We interpret this phenomenon using the information bottleneck principle: the final layer of a deep neural network, activated by the sigmoid or softmax activation functions, causes an information bottleneck, and as a result, only a subset of the task-relevant information is passed on to the output. We first support this argument through a simulated toy experiment and then propose a method to reduce the information bottleneck by removing the last activation function. In addition, we introduce a new pooling method that further encourages the transmission of information from non-discriminative regions to the classification. Our experimental evaluations demonstrate that this simple modification significantly improves the quality of localization maps on both the PASCAL VOC 2012 and MS COCO 2014 datasets, exhibiting a new state-of-the-art performance for weakly supervised semantic segmentation. The code is available at: https://github.com/jbeomlee93/RIB.

## 1   Introduction

Semantic segmentation is the task of recognizing objects in an image using pixel-level allocation of a semantic label. The development of deep neural networks (DNNs) has led to significant advances in semantic segmentation [9, 24]. Training a DNN for semantic segmentation requires a dataset containing a large number of images annotated with pixel-level labels. However, preparing such a dataset requires considerable effort; for example, producing a pixel-level annotation for a single image in the Cityscapes dataset [12] takes more than 90 minutes. This high dependence on pixel-level labels can be alleviated by weakly supervised learning [62, 31, 34, 2].

The objective of weakly supervised semantic segmentation is to train a segmentation network with weak annotations, which provide less information about the location of a target object than pixel-level labels, but are cheaper to obtain. Weak supervision takes the form of scribbles [52], bounding boxes [50, 29, 34], or image-level class labels [31, 3, 2]. In this study, we focus on image-level class labels, because they are the cheapest and most popular option of weak supervision. Most methods that use class labels [31, 32, 55, 2, 7] generate pseudo ground truths for training a segmentation network using localization (attribution) maps obtained from a trained classifier, such as a CAM [66] or a Grad-CAM [48]. However, these maps identify only small regions of a target object that are discriminative for the classification [31, 2, 5] and do not identify the entire region occupied by the object, making the attribution maps unsuitable for training a semantic segmentation network. We interpret this phenomenon using the *information bottleneck* principle [54, 49, 46, 53].

---

[*]Correspondence to: Sungroh Yoon <sryoon@snu.ac.kr>.

35th Conference on Neural Information Processing Systems (NeurIPS 2021).

The information bottleneck theory analyzes the information flow through sequential DNN layers: information regarding the input is compressed as much as possible as it passes through the layers of a DNN, while preserving as much of the task-relevant information as possible. This is advantageous for obtaining optimal representations for classification [15, 1] but is disadvantageous when applying the attribution maps from the resulting classifier to weakly supervised semantic segmentation. The information bottleneck prevents the non-discriminative information of the target object from being considered in the classification logit, and thus, the attribution maps focus on only the small discriminative regions of the target object.

We argue that the information bottleneck becomes prominent in the final layer of the DNN due to the use of the double-sided saturating activation function therein (*e.g.,* sigmoid, softmax). We propose a method to reduce this information bottleneck in the final layer of the DNN by retraining the DNN without the last activation function. Additionally, we introduce a new pooling method that allows more information embedded in non-discriminative features, rather than discriminative features, to be processed in the last layer of a DNN. As a result, the attribution maps of the classifier obtained by our method contain more information on the target object.

The main contributions of this study are summarized as follows. First, we highlight that the information bottleneck occurs mostly in the final layer of the DNN, which causes the attribution maps obtained from a trained classifier to restrict their focus to small discriminative regions of the target object. Second, we propose a method to reduce this information bottleneck by simply modifying the existing training scheme. Third, our method significantly improves the quality of the localization maps obtained from a trained classifier, exhibiting a new state-of-the-art performance on the PASCAL VOC 2012 and MS COCO 2014 datasets for weakly supervised semantic segmentation.

## 2 Preliminaries

### 2.1 Information Bottleneck

Given two random variables $X$ and $Y$, the mutual information $\mathcal{I}(X;Y)$ quantifies the mutual dependence between the two variables. Data processing inequality (DPI) [13] infers that any three variables $X$, $Y$, and $Z$ that form a Markov Chain $X \to Y \to Z$ satisfy $\mathcal{I}(X;Y) \geq \mathcal{I}(X;Z)$. Each layer in a DNN processes the input only from the previous layer, which means that the DNN layers form a Markov chain. Therefore, the information flow through these layers can be represented using DPI. More specifically, when an $L-$layered DNN generates an output $\hat{Y}$ from a given input $X$ through intermediate features $T_l$ ($1 \leq l \leq L$), it forms a Markov Chain $X \to T_1 \to \cdots \to T_L \to \hat{Y}$, and the corresponding DPI chain can be expressed as follows:

$$\mathcal{I}(X;T_1) \geq \mathcal{I}(X;T_2) \geq \cdots \geq \mathcal{I}(X;T_{L-1}) \geq \mathcal{I}(X;T_L) \geq \mathcal{I}(X;\hat{Y}). \tag{1}$$

This implies that the information regarding the input $X$ is compressed as it passes through the layers of the DNN.

Training a classification network can be interpreted as extracting maximally compressed features of the input that preserve as much information as possible for classification; such features are commonly referred to as minimum sufficient features (*i.e.,* discriminative information). The minimum sufficient features (optimal representations $T^*$) can be obtained by the *information bottleneck* trade-off between the mutual information of $X$ and $T$ (compression), and that of $T$ and $Y$ (classification) [54, 15]. In other words, $T^* = \mathrm{argmin}_T \ \mathcal{I}(X;T) - \beta\mathcal{I}(T;Y)$, where $\beta \geq 0$ is a Lagrange multiplier.

Shwartz-Ziv *et al.* [49] observe a *compression phase* in the process of finding the optimal representation $T^*$: when observing $\mathcal{I}(X,T_l)$ for a fixed $l$, $\mathcal{I}(X,T_l)$ steadily increases during the first few epochs, but decreases in the later epochs. Saxe *et al.* [46] argue that the compression phase is mainly observed in DNNs equipped with double-sided saturating non-linearities (*e.g.,* tanh and sigmoid), and is not observed in those equipped with single-sided saturating non-linearities (*e.g.,* ReLU). This implies that DNNs with single-sided saturating non-linearities experience less information bottleneck than those with double-sided saturating non-linearities. This can also be understood in terms of gradient saturation in the double-sided saturating non-linearities: the gradient of those non-linearities with respect to an input above a certain value saturates close to zero [8]. Therefore, features above a certain value will have near-zero gradients during the back-propagation process and be restricted from additionally contributing to the classification.

## 2.2 Class Activation Mapping

A class activation map (CAM) [66] identifies regions of an image focused by a classifier. The CAM is based on a convolutional neural network with global average pooling (GAP) before its final classification layer. This is realized by considering the class-specific contribution of each channel of the last feature map to the classification score. Given a classifier parameterized by $\theta = \{\theta_f, w\}$ where $f(\cdot; \theta_f)$ is the feature extractor prior to GAP, and $w$ is the weight of the final classification layer, a CAM of the class $c$ is obtained from an image $x$ as follows:

$$\mathtt{CAM}(x; \theta) = \frac{\mathbf{w}_c^\mathsf{T} f(x; \theta_f)}{\max \mathbf{w}_c^\mathsf{T} f(x; \theta_f)}, \tag{2}$$

where $\max(\cdot)$ is the maximum value over the spatial locations for normalization.

## 2.3 Related Work

**Weakly Supervised Semantic Segmentation:** Weakly supervised semantic segmentation methods with image-level class labels first construct an initial seed by obtaining a high-quality localization map from a trained classifier. Erasure methods [56, 37, 22] prevent a classifier from only focusing on the discriminative parts of objects by feeding the image from which the discriminative regions have been erased to the classifier. Several contexts of a target object can be considered by combining multiple attribution maps obtained from differently dilated convolutions [31, 57] or from the different layers of a DNN [35]. Diverse images of a target class can be utilized by considering cross-image semantic similarities and differences [17, 51]. Zhang *et al.* [62] analyze the causalities among images, contexts, and class labels and propose CONTA to remove the confounding bias in the classification. Because the localization maps obtained by a classification network cannot accurately represent the boundary of a target object, the initial seed obtained by the above methods is refined using subsequent boundary refinement techniques such as PSA [3] and IRN [2].

**Information Bottleneck:** Tishby *et al.* [54] and Shwartz *et al.* [49] use the information bottleneck theory to analyze the inner workings of a DNN. The concept of the information bottleneck has been employed in many research fields. Dubois *et al.* [15] and Achille *et al.* [1] utilize the information bottleneck to obtain optimal representations from DNNs. DICE [45] is proposed for a model ensemble with the information bottleneck principle: it aims to reduce not only the unnecessary mutual information between features and inputs but also the redundant information shared between features produced by separately trained DNNs. Jeon *et al.* [26] study the disentangled representation learning of a generative model [19] using the information bottleneck principle. Yin *et al.* [60] design a regularization objective based on information theory to deal with the memorization problem in meta-learning. The information bottleneck principle can also be adopted to generate the visual saliency map of a classifier. Zhmoginov *et al.* [65] find important regions for the classifier with the information bottleneck trade-off, and Schulz *et al.* [47] restrict the information flow by adding noise to intermediate feature maps and quantify the amount of information contained in the image region.

## 3 Proposed Method

Weakly supervised semantic segmentation methods using class labels produce a pixel-level localization map from a classifier using CAM [66] or Grad-CAM [48]; however, such a map identifies only small discriminative regions of the target object. We analyze this phenomenon with the information bottleneck theory in Section 3.1 and propose RIB, a method to address this problem, in Section 3.2. We then explain how we train a segmentation network with localization maps improved through RIB in Section 3.3.

### 3.1 Motivation

As mentioned in Section 2.1, the DNN layers with double-sided saturating non-linearities have a larger information bottleneck than those with single-sided saturating non-linearities. The intermediate layers of popular DNN architectures (*e.g.,* ResNet [21] and DenseNet [23]) are coupled with the ReLU activation function, which is a single-sided saturating non-linearity. However, the final layer of these networks is activated by a double-sided saturating non-linearity such as sigmoid or softmax, and the class probability $p$ is computed with the final feature map $T_L$ and the final classification

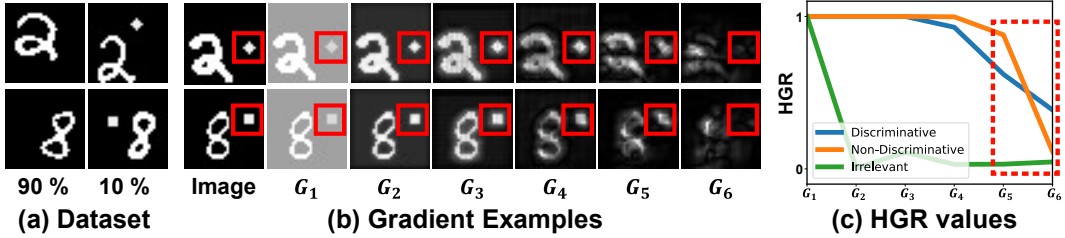

Figure 1: (a) Examples of toy images. (b) Examples of gradient maps $G_k$. (c) Plot of HGR values of $\mathcal{R}_{\text{D}}$, $\mathcal{R}_{\text{ND}}$, and $\mathcal{R}_{\text{BG}}$ for each layer, averaged over 100 images.

layer $w$, *i.e.,* $p = \texttt{sigmoid}(w^{\mathsf{T}}\text{GAP}(T_L))$. Therefore, the final layer parameterized by $w$ has a significant bottleneck, and the amount of information transmitted from the last feature $T_L$ to the actual classification prediction will be limited.

These arguments are analogous to the observations in existing methods. The information plane provided by Saxe *et al.* [46] shows that the compression of information is more noticeable in the final layer than in the other layers. Bae *et al.* [5] observe that although the final feature map of the classifier contains rich information on the target object, the final classification layer filters out most of it; thus, the CAM cannot identify the entire area of the target object. This observation empirically supports the occurrence of the information bottleneck in the final layer of a DNN.

To take a closer look at this phenomenon, we design a toy experiment. We collect images containing the digits '2' or '8' from the MNIST dataset [30]. For only a small subset (10%) of these images, we add a circle (●) and a square (■) to the images containing the digits '2' and '8', respectively, at a random location (see Figure 1(a)). When classifying images into the digits '2' or '8', pixels corresponding to the digit are discriminative regions ($\mathcal{R}_{\text{D}}$), those corresponding to the added circle or square are non-discriminative but class-relevant regions ($\mathcal{R}_{\text{ND}}$), and those corresponding to the background are class-irrelevant regions ($\mathcal{R}_{\text{BG}}$).

We train a neural network with five convolutional layers followed by a final fully connected layer. We obtain the gradient map $G_l$ of each feature $T_l$ with respect to an input image $x$: $G_l = \nabla_x \sum_{u,v} T_l(u, v)$, where $u$ and $v$ are the spatial and channel indices of the feature $T_l$, and for the final classification layer ($l = 6$), $G_6 = \nabla_x y^c$. Because this gradient map indicates the extent to which each pixel of the image affects each feature, it can be used to examine how much information is passed from the input image to the feature maps of successive convolution layers.

We present examples of $G_l$ in Figure 1(b). As an input image passes through the convolution layers, the overall amount of gradient with respect to the input decreases, indicating the occurrence of the information bottleneck. Specifically, the gradient of $\mathcal{R}_{\text{BG}}$ decreases early on ($G_1 \rightarrow G_2$), which implies that the task-irrelevant information is rapidly compressed. From $G_1$ to $G_5$, the gradient in $\mathcal{R}_{\text{D}}$ or $\mathcal{R}_{\text{ND}}$ gradually decreases. However, the decrease in the amount of gradient is prominent in the final layer ($G_5 \rightarrow G_6$), and in particular, the gradients in $\mathcal{R}_{\text{ND}}$ (red boxes) almost disappear. This supports our argument that there is significant information bottleneck in the final layer of a DNN, while also highlighting that the non-discriminative information in $\mathcal{R}_{\text{ND}}$ is particularly compressed.

We analyze this quantitatively. We define the high gradient ratio (HGR) of region $\mathcal{R}$ as the ratio of pixels that have a gradient above 0.3 to the total pixels in region $\mathcal{R}$. HGR quantifies the amount of transmitted information from region $\mathcal{R}$ of an input image to each feature. The trend in the HGR values of each region for each layer is shown in Figure 1(c). The observed trend is analogous to the above empirical observation, once again supporting that significant information bottleneck for $\mathcal{R}_{\text{ND}}$ occurs in the final layer (the red box).

We argue that the information bottleneck causes the localization map obtained from a trained classifier to focus on small regions of the target object. According to Eq. 2, the CAM only includes information that is processed by the final classification weight $w_c$. However, because only a subset of the information in the feature is passed through the final layer $w_c$ due to the information bottleneck, leaving out most of the non-discriminative information, CAM cannot identify the non-discriminative regions of the target object. It is undesirable to use such CAMs to train a semantic segmentation network, for which the entire region of the target object should be identified. Therefore, we aim to bridge the gap between classification and localization by reducing the information bottleneck.

## 3.2 Reducing Information Bottleneck

In Section 3.1, we observed that the information contained in an input image is compressed particularly in the final layer of the DNN, due to the use of the double-sided saturating activation function therein. Therefore, we propose a method to reduce the information bottleneck of the final layer by simply removing the sigmoid or softmax activation function used in the final layer of the DNN. We focus on a multi-class multi-label classifier, which is the default setting for weakly supervised semantic segmentation. Suppose we are given an input image $x$ and the corresponding one-hot class label $t = [t_1, \cdots, t_{\mathcal{C}}]$, where $t_c \in \{0, 1\}$ ($1 \leq c \leq \mathcal{C}$) is an indicator of a class $c$, and $\mathcal{C}$ is the set of all classes. While existing methods use the sigmoid binary cross-entropy (BCE) loss ($\mathcal{L}_{\text{BCE}}$) to train a multi-label classifier, our method replaces it with another loss function $\mathcal{L}_{\text{RIB}}$ that does not rely on the final sigmoid activation function:

$$\mathcal{L}_{\text{BCE}} = -\sum_{c=1}^{\mathcal{C}} t_c \log \texttt{sigmoid}(y^c) + (1 - t_c) \log(1 - \texttt{sigmoid}(y^c)), \ \mathcal{L}_{\text{RIB}} = -\sum_{c=1}^{\mathcal{C}} t_c \min(m, y^c),$$

where $m$ is a margin, and $y^c$ is the classification logit of image $x$.

However, training a classifier with $\mathcal{L}_{\text{RIB}}$ from scratch causes instability in the training because the gradient cannot saturate (please see the Appendix). Therefore, we first train an initial classifier with $\mathcal{L}_{\text{BCE}}$ whose trained weights are denoted by $\theta_0$, and for a given image $x$, we adapt the weights toward a bottleneck-free model of $x$. Specifically, we fine-tune the initial model using $\mathcal{L}_{\text{RIB}}$ computed from $x$ and obtain a model parameterized by $\theta_k$ ($0 < k \leq K$), where $\theta_k = \theta_{k-1} - \lambda \nabla_{\theta_{k-1}} \mathcal{L}_{\text{RIB}}$, and $K$ and $\lambda$ are respectively the total number of iterations and the learning rate for fine-tuning. We name this fine-tuning process RIB. Employing RIB reduces the information bottleneck for $x$, and we can obtain CAMs that identify more regions of the target object, including non-discriminative regions. We repeat the RIB process for all the training images to obtain the CAMs.

However, the model that is adapted to a given image $x$ can be easily over-fitted to $x$. Therefore, to further stabilize the RIB process, we construct a batch of size $B$ for RIB by sampling random $B - 1$ samples other than $x$ at each RIB iteration. Note that for each iteration, $B - 1$ samples are randomly selected, while $x$ is fixed.

**Effectiveness of RIB:** We demonstrate the effectiveness of RIB by applying it to the same classifier as that used for the toy experiments described in Section 3.1. Figure 2 presents (a) examples of $G_6$ and (b) the HGR values for $\mathcal{R}_{\text{D}}$, $\mathcal{R}_{\text{ND}}$, and $\mathcal{R}_{\text{BG}}$ of $G_6$, which showed the most significant information bottleneck, at each RIB iteration. The HGR values are aver-

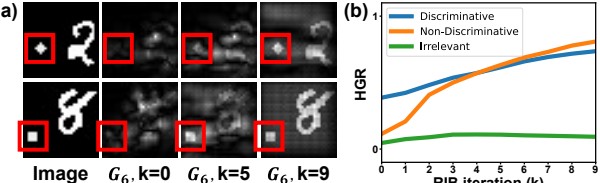

Figure 2: Analysis of $G_6$ for $\mathcal{R}_{\text{D}}$, $\mathcal{R}_{\text{ND}}$, and $\mathcal{R}_{\text{BG}}$ at each RIB iteration.

aged over 100 images. The HGR values of $\mathcal{R}_{\text{BG}}$ remain fairly constant during the RIB process, while the HGR values of $\mathcal{R}_{\text{D}}$ and $\mathcal{R}_{\text{ND}}$ increase significantly. This indicates that the RIB process can indeed reduce the information bottleneck, thereby ensuring that more information corresponding to both $\mathcal{R}_{\text{D}}$ and $\mathcal{R}_{\text{ND}}$ is processed by the final classification layer.

**Limiting the transmission of information from discriminative regions**: Zhang *et al.* [64] showed the relationship between a classification logit $y$ and a CAM, *i.e.,* $y = \text{GAP}(\text{CAM})$. This implies that increasing $y^c$ with RIB also increases the pixel values in the CAM. For a CAM to identify a wider area of the target object, it is important to increase the pixel scores of the non-discriminative regions, rather than the discriminative regions. Therefore, we introduce a new pooling method to the RIB process, so that the features that were previously delivering a small amount of information to the classification logit contribute more to the classification.

We propose a global non-discriminative region pooling (GNDRP). Contrary to GAP which aggregates all the values of the spatial location in the feature map $T_l$, our GNDRP selectively aggregates the values of spatial locations whose CAM scores are below a threshold $\tau$, as follows:

$$\text{GAP}(T_l) = \frac{1}{|\mathcal{U}|} \sum_{u \in \mathcal{U}} T_l(u), \quad \text{GNDRP}(T_l) = \frac{1}{|\mathcal{U}_\tau|} \sum_{u \in \mathcal{U}_\tau} T_l(u), \quad \mathcal{U}_\tau = \{u \in \mathcal{U} \mid \texttt{CAM}(u) \leq \tau\},$$

where $\mathcal{U}$ is a set of all spatial location indices in $T_l$.

| Method | Refinement Method | PASCAL VOC | | | MS COCO | |
|---|---|---|---|---|---|---|
| | | Seed | CRF | Mask | Seed | Mask |
| PSA CVPR '18 [3] | PSA [3] | 48.0 | - | 61.0 | - | - |
| Mixup-CAM BMVC '20 [6] | | 50.1 | - | 61.9 | - | - |
| Chang *et al.* CVPR '20 [7] | | 50.9 | 55.3 | 63.4 | - | - |
| SEAM CVPR '20 [55] | | 55.4 | 56.8 | 63.6 | 25.1[†] | 31.5[†] |
| AdvCAM CVPR '21 [33] | | 55.6 | 62.1 | 68.0 | - | - |
| RIB (Ours) | | **56.5** | **62.9** | **68.6** | - | - |
| IRN CVPR '19 [2] | IRN [2] | 48.8 | 54.3 | 66.3 | 33.5[‡] | 42.9[‡] |
| MBMNet ACMMM '20 [40] | | 50.2 | - | 66.8 | - | - |
| BES ECCV '20 [10] | | 50.4 | - | 67.2 | - | - |
| CONTA NeurIPS '20 [62] | | 48.8 | - | 67.9 | 28.7[†] | 35.2[†] |
| AdvCAM CVPR '21 [33] | | 55.6 | 62.1 | 69.9 | - | - |
| RIB (Ours) | | **56.5** | **62.9** | **70.6** | **36.5**[‡] | **45.6**[‡] |

Table 1: Comparison of the initial seed (Seed), the seed with CRF (CRF), and the pseudo ground truth mask (Mask) on PASCAL VOC and MS COCO training images, in tems of mIoU (%). [†] denotes the results reported by Zhang *et al.* [62], and [‡] denotes the results obtained by us.

Other methods of weakly supervised semantic segmentation also considered new pooling methods other than GAP to obtain better localization maps [4, 27, 44]. The pooling methods introduced in previous works make the classifier focus more on discriminative parts. In contrast, GNDRP excludes highly activated regions, encouraging non-discriminative regions to be further activated.

**Obtaining a final localization map:** We obtain the final localization map $\mathcal{M}$ by aggregating all the CAMs obtained from the classifier at each RIB iteration $k$: $\mathcal{M} = \sum_{0 \leq k \leq K} \texttt{CAM}(x; \theta_k)$.

### 3.3 Weakly Supervised Semantic Segmentation

Because a CAM [66] is obtained from down-sampled intermediate features produced by a classifier, it should be up-sampled to the size of the original image. Therefore, it tends to localize the target object coarsely and cannot represent its exact boundary. Many weakly supervised semantic segmentation methods [7, 6, 55, 62, 40, 31] produce pseudo ground truths by modifying their initial seeds using established seed refinement methods [25, 3, 2, 27, 10]. Similarly, we obtain pseudo ground truths by applying IRN [2], a state-of-the-art seed refinement method, to the coarse map $\mathcal{M}$.

In addition, because an image-level class label is void of any prior regarding the shape of the target object, salient object mask supervision is popularly used in existing methods [59, 31, 22, 38]. Salient object mask supervision can also be applied to our method to refine the pseudo ground truths: when a foreground pixel in a pseudo label is identified as background on this map, or a background pixel is identified as foreground, we ignore such pixels in the training of the segmentation network.

## 4 Experiments

### 4.1 Experimental Setup

**Dataset and evaluation metric:** We evaluated our method quantitatively and qualitatively by conducting experiments on the PASCAL VOC 2012 [16] and the MS COCO 2014 [39] datasets. Following the common practice in weakly supervised semantic segmentation [3, 2, 31, 62], we used the PASCAL VOC 2012 dataset, which is augmented by Hariharan *et al.* [20], containing 10,582 training images with objects from 20 classes. The MS COCO 2014 dataset contains approximately 82K training images containing objects of 80 classes. We evaluated our method on 1,449 validation images and 1,456 test images from the PASCAL VOC 2012 dataset and on 40,504 validation images from the MS COCO 2014 dataset, by calculating the mean intersection-over-union (mIoU) values.

**Reproducibility.** We implemented CAM [66] by following the procedure from Ahn *et al.* [2], which is implemented with the PyTorch framework [43]. We used the ResNet-50 [21] backbone for the classification. We fine-tuned our classifier for $K = 10$ iterations with a learning rate of $8 \times 10^{-6}$ and a batch size of $B = 20$. We set the margin $m$ to 600. For the GNDRP, we set $\tau$ to 0.4. For the final semantic segmentation, we used the PyTorch implementation of DeepLab-v2-ResNet101 offered

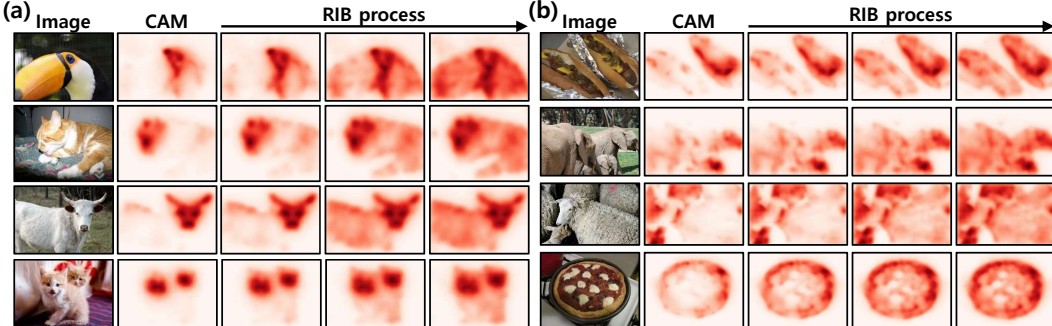

Figure 3: Examples of localization maps obtained during the RIB process for (a) PASCAL VOC 2012 training images and (b) MS COCO 2014 training images.

by [42]. We used an initial model pre-trained on the ImageNet dataset [14]. For the MS COCO 2014 dataset, we cropped the training images with the crop size of $481 \times 481$ rather than $321 \times 321$ used for the PASCAL VOC 2012 dataset, considering the size of the images in this dataset.

## 4.2 Weakly Supervised Semantic Segmentation

### 4.2.1 Quality of the initial seed and pseudo ground truth

**PASCAL VOC 2012 dataset:** In Table 1, we report the mIoU values of the initial seed and pseudo ground truth masks generated from our method and from other recent techniques. Following SEAM [55], we evaluate a range of thresholds to distinguish between the foreground and the background in the map $\mathcal{M}$ and then determine the best initial seeds. Our initial seeds exhibit 7.7%p improvement from the original CAMs, a baseline for comparison, and simultaneously outperform those from the other methods. Note that our initial seeds are better than those of SEAM, which further refines the initial CAM on a pixel-level by considering the relationship between pixels through an auxiliary self-attention module.

We applied a post-processing method based on conditional random field (CRF) [28] for pixel-level refinement of the initial seeds obtained from the method proposed by Chang *et al.* [7], SEAM [55], IRN [2], and our method. On average, applying CRF improved all the seeds by more than 5%p, with the exception of SEAM. CRF improved SEAM by only 1.4%p, and it is reasonable to believe that this unusually small improvement occurred because the self-attention module had already refined the seed from CAM. When the seed produced by our method is refined with CRF, it is 6.1%p better than that from SEAM and consequently outperforms all the recent competitive methods by a large margin.

Additionally, we compare the pseudo ground truth masks obtained after seed refinement with those obtained using other methods. Most of the compared methods use PSA [3] or IRN [2] to refine their initial seeds. For a fair comparison, we generate pseudo ground truth masks using both seed refinement techniques. Table 1 shows that the masks from our method yield an mIoU of 68.6 with PSA [3] and 70.6 with IRN [2], thereby outperforming other methods by a large margin.

**MS COCO 2014 dataset:** Table 1 presents the mIoU values of the initial seed and pseudo ground truth masks obtained by our method and by other recent methods for the MS COCO 2014 dataset. We obtained the results of IRN [2] using the official code to set the baseline performance. Our method improved the initial seed and pseudo ground truth masks of our baseline IRN [2], by mIoU margins of 3.0%p and 2.7%p, respectively.

Figure 3 illustrates localization maps gradually refined by the RIB process for the PASCAL VOC 2012 and the MS COCO 2014 datasets. More samples are shown in the Appendix.

### 4.2.2 Performance of weakly supervised semantic segmentation

**PASCAL VOC 2012 dataset:** Table 2 presents the mIoU values of the segmentation maps on PASCAL VOC 2012 validation and test images, predicted by our method and other recently introduced weakly supervised semantic segmentation methods, which use bounding box labels or image-level class labels. All the results in Table 2 were obtained using a ResNet-based backbone [21]. Our method achieves mIoU values of 68.3 and 68.6 for the validation and test images, respectively, on

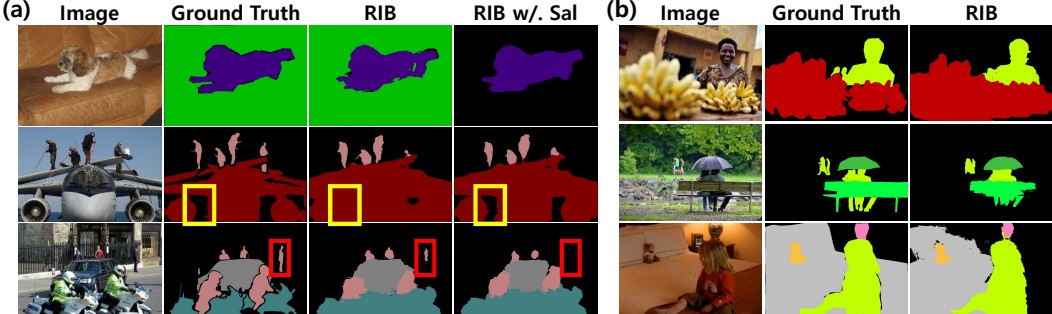

Figure 4: Examples of predicted segmentation masks from IRN [2] and our method for (a) PASCAL VOC 2012 validation images and (b) MS COCO 2014 validation images.

| Method | val | test |
|---|---|---|
| Supervision: Bounding box labels | | |
| Song *et al.* CVPR '19 [50] | 70.2 | - |
| BBAM CVPR '21 [34] | 73.7 | 73.7 |
| Supervision: Image class labels | | |
| IRN CVPR '19 [2] | 63.5 | 64.8 |
| SEAM CVPR '20 [55] | 64.5 | 65.7 |
| BES ECCV '20 [10] | 65.7 | 66.6 |
| Chang *et al.* CVPR '20 [7] | 66.1 | 65.9 |
| RRM AAAI '20 [61] | 66.3 | 66.5 |
| CONTA NeurIPS '20 [62] | 66.1 | 66.7 |
| RIB (Ours) | **68.3** | **68.6** |

Table 2: Comparison of semantic segmentation performance on PASCAL VOC 2012 validation and test images.

| Method | Sup. | val | test |
|---|---|---|---|
| SeeNet NeurIPS '18 [22] | $\mathcal{S}$ | 63.1 | 62.8 |
| FickleNet CVPR '19 [31] | $\mathcal{S}$ | 64.9 | 65.3 |
| CIAN AAAI '20 [17] | $\mathcal{S}$ | 64.3 | 65.3 |
| Zhang *et al.* ECCV '20 [63] | $\mathcal{S}$ | 66.6 | 66.7 |
| Fan *et al.* ECCV '20 [18] | $\mathcal{S}$ | 67.2 | 66.7 |
| Sun *et al.* ECCV '20 [51] | $\mathcal{S}$ | 66.2 | 66.9 |
| LIID TPAMI '20 [41] | $\mathcal{S}_I$ | 66.5 | 67.5 |
| Li *et al.* AAAI '21 [38] | $\mathcal{S}$ | 68.2 | 68.5 |
| Yao *et al.* CVPR '21 [59] | $\mathcal{S}$ | 68.3 | 68.5 |
| RIB (Ours) | $\mathcal{S}$ | **70.2** | **70.0** |

Table 3: Comparison of semantic segmentation performance on PASCAL VOC 2012 validation and test images using explicit localization cues. $\mathcal{S}$: salient object, $\mathcal{S}_I$: salient instance.

the PASCAL VOC 2012 semantic segmentation benchmark, outperforming all the methods that use image-level class labels as weak supervision. In particular, our method outperforms CONTA [62], the best-performing method among our competitors, achieving an mIoU value of 66.1. However, CONTA depends on SEAM [55], which is known to outperform IRN [2]. When CONTA was implemented with IRN for a fairer comparison with our method, its mIoU value decreased to 65.3, which our method surpasses by 3.0%p.

Table 3 compares our method with other recent methods using additional salient object supervision. We utilized salient object supervision used by Li *et al.* [38] and Yao *et al.* [59]. Our method achieves mIoU values of 70.2 and 70.0 for the validation and test images, respectively, outperforming all the recently introduced methods under the same level of supervision.

Figure 4(a) shows examples of predicted segmentation maps by our method with and without saliency supervision. The boundary information provided by saliency supervision allows our method to produce a more precise boundary (yellow boxes). However, the non-salient objects in an image are often ignored when using saliency supervision, while RIB successfully identifies them (*e.g.*, a 'sofa' in the first column and 'person' in red boxes in Figure 4(a)). This empirical finding inspires a potential future work that can simultaneously identify a precise boundary and non-salient objects.

**MS COCO 2014 dataset:** Table 4 compares our method with other recent methods on MS COCO 2014 validation images. Our method achieves an improvement of 2.4%p in terms of the mIoU score compared with our baseline IRN [2], and outperforms the other recent competitive methods [11, 62, 58] by a large margin. In the comparison with CONTA [62], the result of IRN reported in CONTA [62] differs from the one we obtained. Therefore, we compare relative improvements: CONTA achieves a 0.8%p improvement compared with

| Method | Backbone | mIoU |
|---|---|---|
| ADL TPAMI '20 [11] | VGG16 | 30.8 |
| CONTA NeurIPS '20 [62] | ResNet50 | 33.4 |
| Yao *et al.* Access '20 [58] | VGG16 | 33.6 |
| IRN CVPR '19 [2] | ResNet101 | 41.4 |
| RIB (Ours) | ResNet101 | 43.8 |

Table 4: Comparison of semantic segmentation on MS COCO validation images.

| Fine-tuning | Seed | | $m$ | $\lambda$ | Seed | | Method | Seed |
|---|---|---|---|---|---|---|---|---|
| Init. | 48.8 | | 300 | $8 \times 10^{-6}$ | 54.0 | | CAM | 48.8 |
| BCE w/. Tanh | 49.7 | | 600 | $5 \times 10^{-6}$ | 54.9 | | RIB-GAP | 54.8 |
| BCE w/. Sigmoid | 50.5 | | 600 | $8 \times 10^{-6}$ | **56.5** | | RIB-GNDRP ($\tau$=0.3) | 55.8 |
| BCE w/. Softsign | 50.9 | | 600 | $1 \times 10^{-5}$ | 56.0 | | RIB-GNDRP ($\tau$=0.4) | **56.5** |
| $\mathcal{L}_{\text{RIB}}$ | **56.5** | | 1000 | $8 \times 10^{-6}$ | 55.9 | | RIB-GNDRP ($\tau$=0.5) | 56.0 |
| (a) | | | (b) | | | | (c) | |

Table 5: Comparison of mIoU scores of the initial seed (a) with different activation functions for the final layer, (b) with different values of $m$ and $\lambda$, and (c) with different values of $\tau$.

IRN (32.6 $\rightarrow$ 33.4), whereas our method achieves 2.4%p (41.4 $\rightarrow$ 43.8). Figure 4(b) presents examples of predicted segmentation maps by our method for the MS COCO 2014 validation images.

## 4.3 Ablative Studies

In this section, we analyze our method through various ablation studies conducted on the PASCAL VOC 2012 dataset to provide more information about the effectiveness of each component of our method.

**Influence of the total number of RIB iterations $K$:** We analyze the influence of the iteration number $K$ on the effectiveness the RIB process. Figure 5 shows the mIoU score of the initial seed obtained by our baseline CAM, and that of each iteration of the RIB process with GAP or GNDRP. As the RIB process progresses, the localization map is significantly improved, regardless of the pooling method. However, the increase in the performance of RIB with GAP is limited, and even slightly decreases in later iterations ($K > 5$). This is because GAP allows features that have already delivered sufficient information to the classification to become even more involved in the classification. Because our proposed GNDRP limits the increase in the

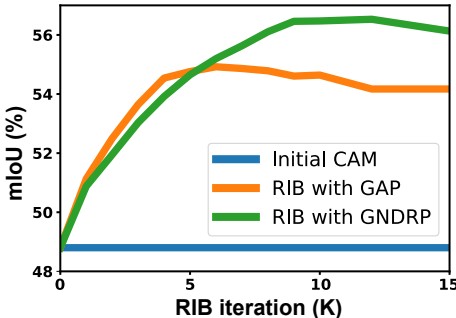

Figure 5: Analysis of RIB with GAP or GNDRP in terms of mIoU of the initial seed.

contribution of these discriminative regions to the classification, RIB with GNDRP can effectively allow non-discriminative information to be more involved in the classification, resulting in a better localization map in later iterations. We observe that changing the value of $K$ to be larger than 10 (even 20) produces less than 0.8%p drop in mIoU, suggesting that it is not difficult to select a good value of $K$.

**Fine-tuning with $\mathcal{L}_{\text{RIB}}$:** To verify the effectiveness of $\mathcal{L}_{\text{RIB}}$, we fine-tune a model using the BCE loss with various double-sided saturating activation functions. Table 5 (a) shows the mIoU scores of the initial seeds, obtained from a model fine-tuned by the BCE loss with sigmoid, tanh, and softsign activations, and our $\mathcal{L}_{\text{RIB}}$. We adjusted the output of tanh and softsign to have a value between zero and one through the affine transform. Fine-tuning using the BCE loss with double-sided saturating activations improves the initial seed to some extent, which demonstrates the effectiveness of per-sample adaptation; however, their performance improvement is limited due to the remaining information bottleneck. Note that the softsign activation function provides better localization maps than tanh and sigmoid. We believe this is because the gradients from softsign reach zero at a higher value compared with the others (please see the Appendix), and consequently, softsign has less information bottleneck. Our $\mathcal{L}_{\text{RIB}}$ effectively addresses the information bottleneck and achieves the best performance.

**Analysis of the sensitivity to hyper-parameters:** We analyze the sensitivity of the mIoU of the initial seed to the hyper-parameters involved in the RIB process. Table 5 (b) presents the mIoU values of the initial seed obtained using different combinations of values for the margin $m$ and the learning rate $\lambda$. Overall, a slightly lower performance is observed when the strength of the RIB process is weakened by small values of $m$ and $\lambda$. For sufficiently large $m$ and $\lambda$, the performance of the RIB process is competitive. Table 5 (c) analyzes the influence of the threshold $\tau$ involved in the GNDRP. Increasing $\tau$ from 0.3 to 0.5 results in less than 1%p change in the mIoU, and thus, we conclude that the RIB process is robust against the changes in $\tau$.

| Method | Prec. | Recall | F1-score |
|---|---|---|---|
| IRN $_{CVPR\ '19}$ [2] | 66.0 | 66.4 | 66.2 |
| Chang *et al.* $_{CVPR\ '20}$ [7] | 61.0 | 77.2 | 68.1 |
| SEAM $_{CVPR\ '20}$ [55] | 66.8 | 76.8 | 71.5 |
| RIB (Ours) | 67.3 | 78.9 | 72.6 |

Table 6: Comparison of precision (Prec.), recall, and F1-score on PASCAL VOC 2012 train images.

| Method | Boat | Train |
|---|---|---|
| IRN $_{CVPR\ '19}$ [2] | 35.3 | 51.3 |
| Chang *et al.* $_{CVPR\ '20}$ [7] | 34.1 | 53.1 |
| SEAM $_{CVPR\ '20}$ [55] | 31.0 | 54.2 |
| RIB (Ours) | 38.8 | 55.1 |

Table 7: mIoU (%) of the initial seed for 'boat' and 'train' classes.

### 4.4 Analysis of Spurious Correlation

In a natural image, the target object and the background can be spuriously correlated when objects of a certain class primarily occur together in a specific context [36, 62] (*e.g.,* a boat on the sea and a train on the rail). Since image-level class labels do not provide an explicit localization cue of the target object, the classifier trained with these labels is vulnerable to spurious correlation. The localization map obtained from the classifier may also highlight the spuriously correlated background, which reduces precision. This is a long-standing problem that is commonly found in weakly supervised semantic segmentation and object localization.

RIB may also activate some portion of the spurious background. However, we find that the amount of discovered areas correctly belonging to the foreground is noticeably more significant by comparing the precision, recall, and F1-score of our method with those of other recent methods in Table 6. Chang *et al.* [7] achieve high recall but experience a large decrease in precision. SEAM [55] avoids this loss of precision with the help of pixel-level refinement implemented with an additional module mentioned in Section 4.2.1. Our method improves precision as well as recall of our baseline IRN [2] without an external module.

To further analyze the spuriously correlated background, we present class-wise seed improvement by our method and other recent methods. We select two representative classes, 'boat' and 'train', which are known to have the background that is spuriously correlated with the foreground (a boat on the sea and a train on the rail). Table 7 shows that RIB can improve localization quality (mIoU) even for classes known to have a spurious foreground-background correlation.

## 5 Conclusions

In this study, we addressed the major challenge in weakly supervised semantic segmentation with image-level class labels. Through the information bottleneck principle, we first analyzed why the localization map obtained from a classifier identifies only a small region of the target object. Our analysis highlighted that the amount of information delivered from an input image to the output classification is largely determined by the final layer of the DNN. We then developed a method to reduce the information bottleneck through two simple modifications to the existing training scheme: the removal of the final non-linear activation function in the DNN and the introduction of a new pooling method. Our method significantly improved the localization maps obtained from a classifier, exhibiting a new state-of-the-art performance on the PASCAL VOC 2012 and MS COCO 2014 datasets.

**Societal implications:** This work may have the following societal impacts. Object segmentation without the need for pixel-level annotation will save resources for research and commercial development. It is particularly useful in fields such as medicine, where expert annotation is costly. However, there are companies that provide annotations for images as a part of their services. If the dependence of DNNs on labels is reduced by weakly supervised learning, these companies may need to change their business models.

## 6 Acknowledgements

This work was supported by Institute of Information & communications Technology Planning & Evaluation (IITP) grant funded by the Korea government(MSIT) [NO.2021-0-01343, Artificial Intelligence Graduate School Program (Seoul National University)], the National Research Foundation of Korea (NRF) grant funded by the Korea government (MSIT) [2018R1A2B3001628], AIRS Company in Hyundai Motor and Kia through HMC/KIA-SNU AI Consortium Fund, and the Brain Korea 21 Plus Project in 2021.

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
