# Appendix for "Reducing Information Bottleneck for Weakly Supervised Semantic Segmentation"

## S1  Implementation Details

**Optimization details for semantic segmentation:** For the PASCAL VOC 2012 dataset, we set the batch size to 10, the number of training iterations to 30K, and the learning rate to $2.0 \times 10^{-4}$. For the MS COCO 2014 dataset, we set the batch size to 10, the number of training iterations to 100K, and the learning rate to $2.5 \times 10^{-4}$. We use a balanced cross-entropy loss for training a segmentation network, as done in the previous methods [5, 4, 7]. We use a similar re-training technique to the previous methods [4, 5, 6, 12, 11]. In the map obtained after executing random walk of IRN [1], we define pixels with a value greater than 0.3 as foreground and pixels with values less than 0.2 as background, and ignore the remaining pixels ($\mathcal{P}_{\text{ignore}}$) in the initial segmentation training process. We fill the labels of $\mathcal{P}_{\text{ignore}}$ using the segmentation maps predicted by the initially trained segmentation network, and train the network again with all the pseudo labels filled in. Without the re-training technique, our method obtains 67.83 mIoU, which outperforms all the methods presented in Table 2 by a large margin. Note that we do not employ the re-training technique for RIB with saliency and for the MS COCO dataset.

**Computation Resources:** Our experiments were performed on four NVIDIA Quadro RTX 8000 GPUs. The RIB process for Pascal VOC train split (1,464 images) takes 32 minutes and 43 minutes on four NVIDIA Quadro RTX 8000 and four NVIDIA Tesla V100 GPUs, respectively.

## S2  Additional Analysis

**Training a classifier with $\mathcal{L}_{\text{RIB}}$ from scratch:** In Section 3.2, we argue that training a classifier with $\mathcal{L}_{\text{RIB}}$ from scratch causes instability in training. We support this with loss curves obtained with different values of the learning rate in Figure S1(a). Since the gradient of the loss does not saturate, the loss diverges to $-\infty$ after a few iterations.

**Different double-sided saturating activation functions:** In Section 4.3, we fine-tuned the initial model with the BCE loss with tanh, sigmoid, and softsign activations. As shown in Figure S1(b), the tanh activation saturates the fastest, and the softsign activation shows the most linear-like behavior, indicating that the information bottleneck is largest in tanh and smallest in softsign. This is supported by our experimental results in Table 5(a) of the main paper: the fine-tuning process was effective in the order of softsign, sigmoid, and tanh.

**Error bars:** We repeat our RIB process five times to investigate the sensitivity of the initial seed to the random seeds. The obtained mIoU score is $56.44 \pm 0.05$.

**Sensitivity of a batch size $B$:** We analyze the sensitivity of the mIoU of the initial seed to the values of a batch size $B$. Table S1 shows the mIoU scores of the initial seed for the PASCAL VOC dataset for different values of $B$. The RIB process is more effective when using additional $B - 1$ samples other than the target image $x$ to construct a batch than when using only $x$ as a batch ($B = 1$). In addition, the performance starts to saturate above a certain value of $B$, which shows that selecting a good value of $B$ is rather straightforward.

**More examples:** Figures S2 and S3 present examples of localization maps gradually refined by the RIB process for the PASCAL VOC and the MS COCO datasets. Figure S4 presents examples of segmentation maps predicted by our method.

**Per-class mIoU scores:** We present the per-class mIoU of our method and other recently introduced methods for the PASCAL VOC dataset (Table S2) and the MS COCO dataset (Table S3).

35th Conference on Neural Information Processing Systems (NeurIPS 2021).

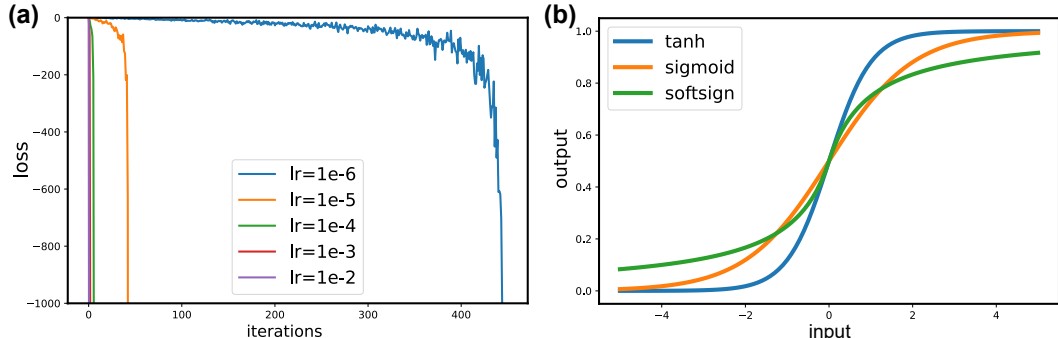

Figure S1: (a) Loss curves with different values of the learning rate. (b) Visualization of tanh, sigmoid, and softsign activations.

Table S1: Comparison of mIoU scores of the initial seed with different values of $B$.

| $B$ | 1 | 5 | 10 | 15 | 20 | 25 | 30 |
|---|---|---|---|---|---|---|---|
| mIoU | 55.3 | 55.5 | 55.8 | 56.2 | 56.5 | 56.6 | 56.7 |

Table S2: Comparison of per-class mIoU scores for the PASCAL VOC dataset.

| | bkg | aero | bike | bird | boat | bottle | bus | car | cat | chair | cow | table | dog | horse | motor | person | plant | sheep | sofa | train | tv | mIOU |
|---|---|---|---|---|---|---|---|---|---|---|---|---|---|---|---|---|---|---|---|---|---|---|
| Results on PASCAL VOC 2012 validation images: | | | | | | | | | | | | | | | | | | | | | | |
| PSA [2] | 88.2 | 68.2 | 30.6 | 81.1 | 49.6 | 61.0 | 77.8 | 66.1 | 75.1 | 29.0 | 66.0 | 40.2 | 80.4 | 62.0 | 70.4 | 73.7 | 42.5 | 70.7 | 42.6 | 68.1 | 51.6 | 61.7 |
| CIAN [3] | 88.2 | 79.5 | 32.6 | 75.7 | 56.8 | 72.1 | 85.3 | 72.9 | 81.7 | 27.6 | 73.3 | 39.8 | 76.4 | 77.0 | 74.9 | 66.8 | 46.6 | 81.0 | 29.1 | 60.4 | 53.3 | 64.3 |
| SEAM [10] | 88.8 | 68.5 | 33.3 | 85.7 | 40.4 | 67.3 | 78.9 | 76.3 | 81.9 | 29.1 | 75.5 | 48.1 | 79.9 | 73.8 | 71.4 | 75.2 | 48.9 | 79.8 | 40.9 | 58.2 | 53.0 | 64.5 |
| FickleNet [5] | 89.5 | 76.6 | 32.6 | 74.6 | 51.5 | 71.1 | 83.4 | 74.4 | 83.6 | 24.1 | 73.4 | 47.4 | 78.2 | 74.0 | 68.8 | 73.2 | 47.8 | 79.9 | 37.0 | 57.3 | 64.6 | 64.9 |
| SSDD [9] | 89.0 | 62.5 | 28.9 | 83.7 | 52.9 | 59.5 | 77.6 | 73.7 | 87.0 | 34.0 | 83.7 | 47.6 | 84.1 | 77.0 | 73.9 | 69.6 | 29.8 | 84.0 | 43.2 | 68.0 | 53.4 | 64.9 |
| BBAM [8] | 92.7 | 80.6 | 33.8 | 83.7 | 64.9 | 75.5 | 91.3 | 80.4 | 88.3 | 37.0 | 83.3 | 62.5 | 84.6 | 80.8 | 74.7 | 80.0 | 61.6 | 84.5 | 48.6 | 85.8 | 71.8 | 73.7 |
| RIB (Ours) | 90.3 | 76.2 | 33.7 | 82.5 | 64.9 | 73.1 | 88.4 | 78.6 | 88.7 | 32.3 | 80.1 | 37.5 | 83.6 | 79.7 | 75.8 | 71.8 | 47.5 | 84.3 | 44.6 | 65.9 | 54.9 | 68.3 |
| RIB–Sal (Ours) | 91.7 | 85.2 | 37.4 | 80.4 | 69.5 | 72.8 | 89.2 | 81.9 | 89.7 | 29.7 | 84.2 | 30.8 | 85.5 | 84.1 | 79.5 | 75.8 | 52.4 | 83.5 | 38.2 | 74.2 | 59.3 | 70.2 |
| Results on PASCAL VOC 2012 test images: | | | | | | | | | | | | | | | | | | | | | | |
| PSA [2] | 89.1 | 70.6 | 31.6 | 77.2 | 42.2 | 68.9 | 79.1 | 66.5 | 74.9 | 29.6 | 68.7 | 56.1 | 82.1 | 64.8 | 78.6 | 73.5 | 50.8 | 70.7 | 47.7 | 63.9 | 51.1 | 63.7 |
| FickleNet [5] | 90.3 | 77.0 | 35.2 | 76.0 | 54.2 | 64.3 | 76.6 | 76.1 | 80.2 | 25.7 | 68.6 | 50.2 | 74.6 | 71.8 | 78.3 | 69.5 | 53.8 | 76.5 | 41.8 | 70.0 | 54.2 | 65.3 |
| SSDD [9] | 89.0 | 62.5 | 28.9 | 83.7 | 52.9 | 59.5 | 77.6 | 73.7 | 87.0 | 34.0 | 83.7 | 47.6 | 84.1 | 77.0 | 73.9 | 69.6 | 29.8 | 84.0 | 43.2 | 68.0 | 53.4 | 64.9 |
| BBAM [8] | 92.8 | 83.5 | 33.4 | 88.9 | 61.8 | 72.8 | 90.3 | 83.5 | 87.6 | 34.7 | 82.9 | 66.1 | 83.9 | 81.1 | 78.3 | 77.4 | 55.2 | 86.7 | 58.5 | 81.5 | 66.4 | 73.7 |
| RIB (Ours)[1] | 90.4 | 80.5 | 32.8 | 84.9 | 59.4 | 69.3 | 87.2 | 83.5 | 88.3 | 31.1 | 80.4 | 44.0 | 84.4 | 82.3 | 80.9 | 70.7 | 43.5 | 84.9 | 55.9 | 59.0 | 47.3 | 68.6 |
| RIB–Sal (Ours)[2] | 91.8 | 89.4 | 37.1 | 84.7 | 56.7 | 69.2 | 89.6 | 84.0 | 89.8 | 24.6 | 81.3 | 37.9 | 85.4 | 84.5 | 81.1 | 75.5 | 50.7 | 85.9 | 44.7 | 71.9 | 54.0 | 70.0 |

Table S3: Comparison of per-class mIoU scores for the MS COCO dataset.

| Class | IRN | Ours | Class | IRN | Ours | Class | IRN | Ours | Class | IRN | Ours | Class | IRN | Ours |
|---|---|---|---|---|---|---|---|---|---|---|---|---|---|---|
| background | 80.5 | 82.0 | dog | 56.2 | 63.5 | kite | 28.8 | 37.1 | broccoli | 52.6 | 45.4 | cell phone | 51.6 | 54.1 |
| person | 45.9 | 56.1 | horse | 58.1 | 63.6 | baseball bat | 12.6 | 15.3 | carrot | 37.0 | 34.6 | microwave | 42.7 | 45.2 |
| bicycle | 48.9 | 52.1 | sheep | 64.6 | 69.1 | baseball glove | 7.9 | 8.1 | hot dog | 48.4 | 49.7 | oven | 31.0 | 35.9 |
| car | 31.3 | 43.6 | cow | 63.8 | 68.3 | skateboard | 27.1 | 31.8 | pizza | 55.9 | 58.9 | toaster | 16.4 | 17.8 |
| motorcycle | 64.7 | 67.6 | elephant | 79.3 | 79.5 | surfboard | 40.7 | 29.2 | donut | 50.0 | 53.1 | sink | 33.3 | 33.0 |
| airplane | 62.0 | 61.3 | bear | 74.6 | 76.7 | tennis racket | 49.7 | 48.9 | cake | 38.6 | 40.7 | refrigerator | 40.0 | 46.0 |
| bus | 60.4 | 68.5 | zebra | 79.7 | 80.2 | bottle | 30.9 | 33.1 | chair | 17.7 | 20.6 | book | 29.9 | 31.1 |
| train | 51.1 | 51.3 | giraffe | 72.3 | 74.1 | wine glass | 24.3 | 27.5 | couch | 32.6 | 36.8 | clock | 41.3 | 41.9 |
| truck | 32.2 | 38.1 | backpack | 19.1 | 18.1 | cup | 27.3 | 27.4 | potted plant | 10.5 | 17.0 | vase | 28.4 | 27.5 |
| boat | 36.7 | 42.3 | umbrella | 57.3 | 60.1 | fork | 16.9 | 15.9 | bed | 33.8 | 46.2 | scissors | 41.2 | 41.0 |
| traffic light | 48.7 | 47.8 | handbag | 9.0 | 8.6 | knife | 15.6 | 14.3 | dining table | 6.7 | 11.6 | teddy bear | 56.4 | 62.0 |
| fire hydrant | 74.9 | 73.4 | tie | 24.0 | 28.6 | spoon | 8.4 | 8.2 | toilet | 63.4 | 63.9 | hair drier | 16.2 | 16.7 |
| stop sign | 76.8 | 76.3 | suitcase | 45.2 | 49.2 | bowl | 17.0 | 20.7 | tv | 35.5 | 39.7 | toothbrush | 16.7 | 21.0 |
| parking meter | 67.3 | 68.3 | frisbee | 53.8 | 53.6 | banana | 62.4 | 59.8 | laptop | 39.3 | 48.2 | | | |
| bench | 31.4 | 39.7 | skis | 8.0 | 9.7 | apple | 43.3 | 48.5 | mouse | 27.9 | 22.4 | | | |
| bird | 55.5 | 57.5 | snowboard | 25.5 | 29.4 | sandwich | 37.9 | 36.9 | remote | 41.4 | 38.0 | mean | 41.4 | 43.8 |
| cat | 68.2 | 72.4 | sports ball | 33.6 | 38.0 | orange | 60.1 | 62.5 | keyboard | 52.9 | 50.9 | | | |

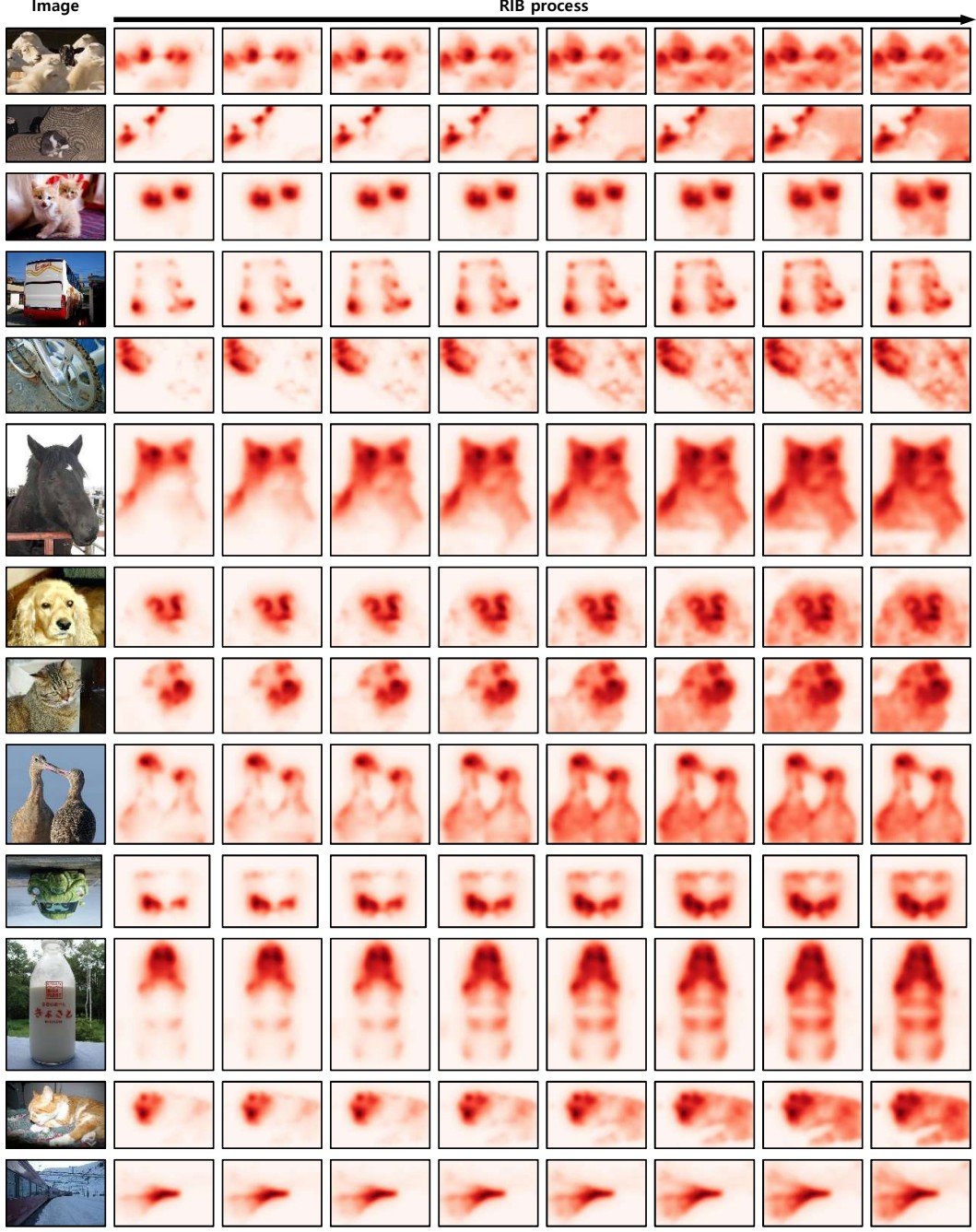

Figure S2: Examples of localization maps obtained during the RIB process for PASCAL VOC training images.

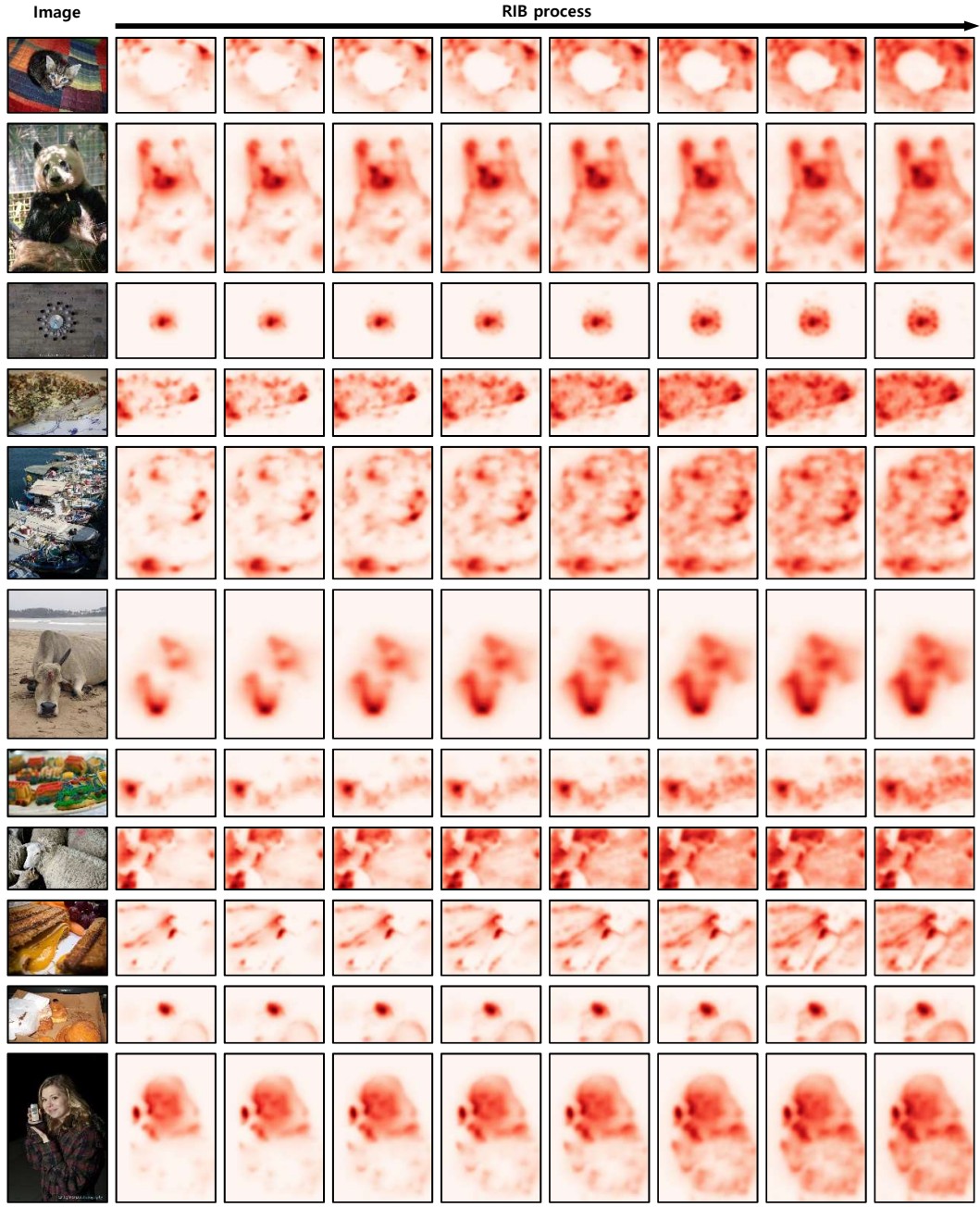

Figure S3: Examples of localization maps obtained during the RIB process for MS COCO training images.

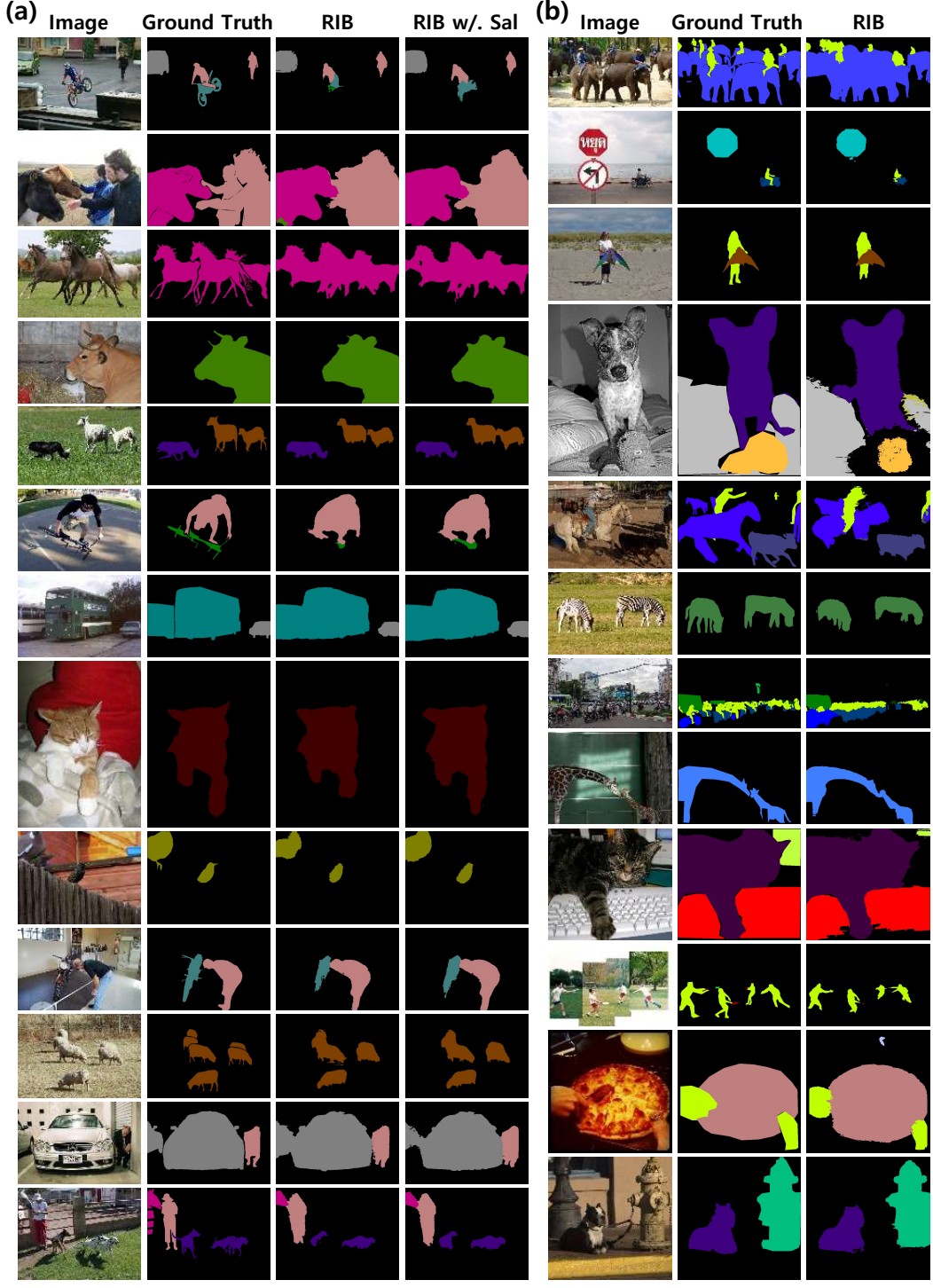

Figure S4: Examples of predicted segmentation masks from IRN [1] and our method for (a) PASCAL VOC 2012 validation images and (b) MS COCO 2014 validation images.