# OpenReview forum: "Reducing Information Bottleneck for Weakly Supervised Semantic Segmentation"
_NeurIPS.cc/2021/Conference — NeurIPS 2021 Poster_

### Official Review · Reviewer_PSWB · 2021-07-14

**Rating:** 7
**Confidence:** 5

**Summary:**

This paper presents a new method for computing class activation maps (CAMs), which has been used as crucial evidence for object detection and segmentation in a weakly supervised learning setting. A well known drawback of CAM and its variants is that they often highlight only small discriminative regions of an object. This paper interprets such a phenomenon through “information bottleneck” and proposes a way to alleviate it accordingly. In short, the proposed method finetune a classification network for generating the original CAM with a different classification loss and a different global pooling layer that allow more features to contribute to classification. The method enhances the quality of the original CAM noticeably and consequently improves performance of a weakly supervised semantic segmentation method relying on CAMs.

The main idea of this work is novel, and useful to alleviate the well-known limitation of the CAM and its variants. Moreover, the efficacy of the proposed method has been demonstrated by extensive experiments. However, the overall pipeline for semantic segmentation depends heavily on an existing method (IRN [2]), and it demands a much larger number of hyper-parameters compared to the original CAM.


**Ethical Concerns:**

No ethical issue has been found.


**Limitations And Societal Impact:**

Limitations of this work are not summarized in a separate section. This reviewer could not find any potential negative societal impact of this work.

**Main Review:**

[Strengths]
- The paper is overall well-written; it is easy to follow in most parts and well-organized. Also, details necessary to reproduce the proposed method are presented in the paper.
- The main idea is novel in that there is no previous work explaining and alleviating the well-known limitation of the CAM and its variants through information bottleneck.
- The efficacy of the proposed method has been demonstrated by extensive experiments on PASCAL VOC and MS-COCO, where weakly supervised semantic segmentation based on the proposed method achieves the state of the art. This reviewer in particular appreciates its strong performance on the challenging MS-COCO dataset.
- Also, the effect of hyper-parameters is investigated by extensive ablation studies.

[Weaknesses / questions]
- The proposed method is applied only on weakly supervised semantic segmentation and its overall pipeline for semantic segmentation relies heavily on IRN [2]; considering this work as a weakly supervised semantic segmentation paper, its contribution looks quite narrow. One way to resolve this issue might be applying the proposed method to other weakly supervised visual recognition tasks, e.g., object localization and instance segmentation.
- The method demands a larger number of hyper-parameters, compared to the original CAM. Although their effects are investigated by ablation studies, this reviewer still wonders how to set the number of RIB iterations, i.e., K, and what happens when K > 10; these analyses are missing in the current manuscript.
- It is not explained why the final localization map is computed as the sum of CAMs over RIB iterations, instead of just choosing the CAM of the last RIB iteration.
- The design of experiments in Section 3.1 is not convincing. In particular, it is not explained at all why the circles and squares should be considered as class-relevant yet non-discriminative regions.
- The authors seem to assume that class-relevant yet non-discriminative regions are always parts of an object, which is not true unfortunately since such regions may include class-specific background stuff (e.g., rails near trains). Thus the proposed RIB finetuning could damage precision while improving recall. In the same context, this reviewer would recommend analyzing, evaluating, and comparing the proposed method with others through precision-recall curves.


**Time Spent Reviewing:**

7

---

> ### Author Response · Authors · 2021-08-10
> **Response to Reviewer PSWB**
>
>
> We appreciate the reviewer's insightful comments and suggestions on our manuscript. Below, we address the reviewer's main concerns.
>
>
> **Q1. Reliance on IRN [2]**
>
> A1. Yes, our method uses the existing seed refinement technique, i.e., IRN [2], but so do
> other state-of-the-art seed generation methods (please refer to Lines 231-234 in the manuscript).
> Most of very recent seed generation works, including those published in NeurIPS and CVPR, have also relied on existing, well-established seed refinement methods.
> With the exception of BES [9], whose primary goal is to propose an improved seed refinement technique,
> all of the methods shown in Table 1 employ either PSA [1] (First 4 methods) or IRN [2] (Others).
> Our contribution comes from the improvement of the initial seed, which results in a better pseudo ground-truth after seed refinement.
> As mentioned in Lines 273-274, the seed produced by our method outperforms all the recent methods by a large margin (over 6%p mIoU).
>
>
> The table below presents the performance of our method and the other recently introduced methods, which depend on IRN to refine their seeds
> (please note that the table below is a re-arranged version of Table 1 in the main paper).
> Our method improves IRN by a large margin (4.3%p mIoU) and outperforms all the methods under the same refinement technique.
>
> |Method|Refine Method|Seed|Mask|
> |------|:---:|:---:|:---:|
> |IRN (CVPR 19)|IRN|48.8|66.3|
> |MBMNET (ACMMM 20)|IRN|50.2|66.8|
> |CONTA (NeurIPS 20)|IRN|48.8|67.9|
> |RIB (Ours)|IRN|56.5|70.6|
>
>
> To further consolidate our contribution for weakly supervised semantic segmentation,
> we present new experimental results by using PSA [1], another popular seed refinement technique.
> The table below presents the performance of our method and the other recently introduced methods, which depend on PSA to refine their seeds.
> RIB improved PSA by 7.6\%p mIoU, and obtained 5.0%p better pseudo ground truth (Mask) than SEAM [51], the best-performing method among our competitors.
>
>
> |Method|Refine Method|Seed|Mask|
> |------|:---:|:---:|:---:|
> |PSA (CVPR 18)|PSA|48.0|61.0|
> |Mixup-CAM (BMVC 20)|PSA|50.1|61.9|
> |Chang et al. (CVPR 20)|PSA|50.9|63.4|
> |SEAM (CVPR 20)|PSA|55.4|63.6|
> |RIB (Ours)|PSA|56.5|68.6|
>
>
>
> **Q2. Extension to the weakly supervised object localization task**
>
> A2.
> As requested by the reviewer, we show that RIB can be extended to weakly supervised object localization (WSOL).
> We conduct experiments on the ImageNet-1K dataset using ResNet-50 backbone.
> We follow the official implementation of [ref1].
> The table below presents Top-1 localization accuracy, MaxBoxAcc@mIoU50 [ref1], MaxBoxAccV2 [ref1] on ImageNet-1K test images, showing that our method is also applicable to WSOL and outperforms other recently introduced WSOL methods.
>
> |Method|Top-1 Loc Acc|MaxBoxAcc@mIoU50|MaxBoxAccV2|
> |----|:---:|:---:|:---:|
> |IVR (ICCV 21) [ref2] |-|-|64.2|
> |Pan et al (CVPR 21) [ref3] |52.7|68.3|-|
> |mLDFR (TIP 21) [ref4] |-|-|65.2|
> |RIB (Ours) |53.7|68.5|66.0|
>
>
> The experimental results presented in A1 and A2 expand the scope of our contribution in two aspects:
>
> - For weakly supervised semantic segmentation, our method outperforms the recent competitive methods by a large margin, regardless of the choice of a seed refinement technique.
>
> - Our method can be generalized to a new task, weakly supervised object localization.
>
> [ref1] Choe et al. "Evaluation for weakly supervised object localization: Protocol, metrics, and datasets." arXiv preprint arXiv:2007.04178 (2020).
>
> [ref2] Kim et al. "Normalization Matters in Weakly Supervised Object Localization." ICCV 2021.
>
> [ref3] Pan et al. "Unveiling the Potential of Structure Preserving for Weakly Supervised Object Localization." CVPR 2021.
>
> [ref4] Wang et al. "Multi-Scale Low-Discriminative Feature Reactivation for Weakly Supervised Object Localization." TIP 2021.
>
>
>
>
> **Q3. Hyper-parameters.**
>
> A3.
>
> - We want to note that our method is not sensitive to hyper-parameters.
> Tables 5(b, c) and Table S1 (in the Appendix) show that our method outperforms most methods over a broad range of hyperparameter values.
> It is thus reasonable to believe that extensive hyper-parameter search is not required to achieve a much superior localization performance to that of CAM.
>
>
> - As requested by the reviewer, we present the localization accuracy (mIoU) when using K > 10, in the table below.
> We observe that changing the value of K to be larger than 10 (even 20) produces less than 0.8%p drop in mIoU, suggesting that it is not difficult to select a
> good value of K.
>
> |K|10|12|15|20|
> |----|:---:|:---:|:---:|:---:|
> |mIoU|56.48|56.53|56.13|55.73|
>
>
> **Q4. Why the final localization map is computed as the sum of CAMs over RIB iterations?**
>
> A4. We compute the final localization map by aggregating CAMs over RIB iterations to suppress the noise occurring in the latter RIB iterations.
> When the CAM from the last RIB iteration is used as the final localization map, we obtain 56.1 mIoU.
> Changing the method of obtaining the final localization map leads to a mere 0.4%p drop in mIoU,
> showing that our method is not highly dependent on the definition of the final localization map.
>
>
>
> **Q5. The design of experiments in Section 3.1.**
>
> A5.
> In the toy experiment presented in Section 3.1, we believe that a circle (or a square) is not spuriously correlated with a digit, but is class-relevant.
> Let us assume that we have classes A and B that frequently co-occur within an image.
> For A and B to be spuriously correlated,
> it should be guaranteed that B is not a part of A (i.e., B belongs in a different class from A).
> To guarantee this, images that contain either class B alone or B and another class C together should exist.
> However, because a circle (or a square) in our toy experiment always appears only with a digit '2' (or a digit '8'),
> our current toy scenario cannot analyze spurious correlation.
>
>
> To take a closer look at the spurious correlation, we design a new toy experiment. We define two different settings.
>   - Spurious correlation setting: We add a square to 70% of the images containing the digit '2', and also add a square to 20% of images containing the digit '8'.
> In this case, the square and the digit '2' frequently occur together,
> and thus they are highly correlated.
> However, this high correlation between the square and the digit '2' is spurious because the square also appears with the digit '8' occasionally.
> This is similar to an example of the spurious correlation between boat and sea: they frequently occur together,
> but the sea also occasionally appears with a bird.
>
>   - Class-relevant setting: We add a square to 70% of the images containing the digit '2', but we do NOT add a square to the images containing the digit '8'. In this case, since the square appears only with the digit '2', we can assume that the square is a part of the digit '2'.
>
>   We train classifiers under each setting and conduct the RIB process for each classifier.
> The table below shows HGR values of $G_6$ during the RIB process in each setting, similar to Figure 2(b).
> In the class-relevant setting, HGR values rapidly increase as the RIB process progresses, meaning that information flow in the square region is greatly increased.
> In contrast, in the spurious background setting, the increase in the HGR values is less prominent.
> It is thus reasonable to believe that our method can handle the spurious background effectively.
>
>
> |Setting|k=0|k=3|k=6|k=9|
> |----|:---:|:---:|:---:|:---:|
> |Class-relevant setting, HGR|0.17|0.35|0.55|0.64|
> |Spurious background setting, HGR|0.03|0.04|0.10|0.21|
>
>
>
>
>
> **Q6. Does RIB damage precision?**
>
> A6. As the reviewer insightfully pointed out, RIB may activate some portion of the spurious background.
> However, we would like to argue that the amount of discovered areas correctly belonging to the foreground is noticeably more significant.
> To support this, we measure precision, recall, and F-score as shown in the table below.
> RIB-GAP improves precision as well as recall of CAM, and GNDRP further improves them.
> In addition, our method improves all the metrics compared to the recent methods.
>
> |Method|Precision|Recall|F-score|
> |----|:---:|:---:|:---:|
> |CAM |66.0|66.4|66.2|
> |Chang et al. (CVPR 20) [6] |61.0|77.2|68.1|
> |SEAM (CVPR 20) [51] |66.8|76.8|71.5|
> |RIB-GAP (Ours) |66.4|77.5|71.5|
> |RIB-GNDRP (Ours) |67.3|78.9|72.6|
>
>
>
> To further analyze the class-specific background, we present class-wise seed improvement by our method and other recent methods.
> We select two representative classes, boat and train, which are known to have class-specific background stuff (boat-sea and train-rail).
> The table below shows that RIB can improve localization quality (mIoU) even for classes known to have a spurious foreground-background correlation.
>
> |Method|Boat|Train|
> |----|:---:|:---:|
> |CAM |35.3|51.3 |
> |Chang et al. (CVPR 20) [6] |34.1|53.1|
> |SEAM (CVPR 20) [51] |31.0|54.2|
> |RIB (Ours) |38.8|55.1|

---

> > ### Comment · Reviewer_PSWB · 2021-08-23
> > **Post-rebuttal review**
> >
> > I greatly appreciate the authors' responses. They address all of my concerns thoroughly, with extensive experiments well supporting the arguments. Also I believe the additional results on other weakly supervised learning tasks make this submission much stronger. Thus I now lean towards acceptance and upgrade my rating accordingly.

---

### Official Review · Reviewer_cUpz · 2021-07-16

**Rating:** 6
**Confidence:** 5

**Summary:**

This paper focuses on the problem of weakly-supervised semantic segmentation, where the goal is to conduct segmentation by mainly using image-level annotation as such weak form of annotations can be collected more efficiently.

Specifically, authors analyze the issue of Information Bottleneck and propose a simple method RIB to reduce it to obtain more complete CAM localization maps. A e a global non-discriminative region pooling (GNDRP) is further applied to enhance the effects of RIB.

Experiments on PASCAL VOC  and MS COCO datasets show the effectiveness of the proposed method.

**Limitations And Societal Impact:**

Not applicable. Authors did not discuss limitations and potential negative societal impacts of the work. See check list 1 (c).

**Main Review:**

Strength:
+ This paper begins with toy experiments to illuminate the problem of information bottleneck, which is intuitive and clear.

+ The ideas of RIB and GNDRP are reasonable and effective.

+ Extensive experiments have been conducted to show the contributions of the proposed method. Parameter analysis has also been conducted to help analyze the behavior of the model.

+ The proposed method shows strong performance when compared with recent state-of-the-art methods.



Concerns and Questions:

- The descriptions and motivations of RIB training method are not clear (line 189-201). For each RIB iteration, is the model updated by the gradient of L_RIB for a fixed image x as well as L_RIB for other N-1 randomly sampled images? Why is it necessary to set several RIB iterations for each specific image x? Why not just conduct RIB loss for images in a batch and fine-tuning the model for several epochs?

- For the details of using GNDRP, is it only applied at the training stage (RIB iteration)? Then at the inference stage of CAM maps, I believe the original GAP layer is still used rather than GNDRP. It that correct? The idea of GNDRP is quite similar to hide-and-seek method operated on feature maps, which is not new.

- For results in Figure 3, are they obtained by only applying RIB iterations (RIB-GAP)? Or they are actually obtained by RIB-GNDRP. If they are from RIB-GAP then it is fine. Otherwise, showing results of RIB-GAP would be more helpful.

- Can the proposed method generalize to Grad-CAM instead of only on CAM? Analysis, discussions and even experimental results regarding this point would help to show the generalization of the method.

**Time Spent Reviewing:**

5

---

> ### Author Response · Authors · 2021-08-10
> **Response to Reviewer cUpz**
>
> We appreciate the reviewer's insightful comments and suggestions on our manuscript. Below, we address the reviewer's main concerns.
>
> **Q1. Descriptions and motivations of RIB training method.**
>
> A1. We are sorry if the reviewer felt that the description of the RIB training method was not as sufficient as it could be.
>
> - Yes, the model is updated by the gradient of $L_{RIB}$ for a fixed (target) image $x$ as well as $L_{RIB}$ for other N-1 randomly sampled images.
>
> - During the RIB process, we adapt the model towards a bottleneck-free model for a single target image, instead of all images, because:
>
>   * First, training the model with $L_{RIB}$ for several epochs causes instability as shown in Figure S1(a) of the Appendix:
> the model diverges within hundreds of iterations.
> Therefore, executing thousands of RIB iterations to obtain the bottleneck-free model for all images is infeasible due to the instability in the fine-tuning process.
> We thus design our method to reduce the information bottleneck for each target image, which can be realized in a few iterations, thereby stabilizing the RIB process.
>
>   * Second, the results in the table below (re-arranged version of Table 5(a) in the main paper) demonstrate that per-sample adaptation is more effective than adapting a model to multiple images.
>   The initial model was previously trained using $L_{BCE}$ through a batch training scheme and thus was adapted to multiple images.
>   As can be seen from the table below, per-sample fine-tuning with the same $L_{BCE}$ and the same activation function improves the initial seed by 1.7%p mIoU.
>   This experimental result supports the effectiveness of per-sample adaptation.
>
> |Fine-tuning|Seed quality (mIoU)|
> |----|:---:|
> |Initial model|48.8|
> |+ Per-sample adaptation with BCE, Sigmoid|50.5|
>
>
> **Q2. Details of using GNDRP.**
>
> A2.
> - Yes, GNDRP is only applied at the fine-tuning stage (RIB iteration).
> As shown in Eq. (2), CAM is only dependent on the feature map $f(x)$ before the pooling layer and the weights of the last layer $w_c$,
> and thus, no pooling layer is involved in the process of generating CAMs.
>
>
> - We assume that by "Hide-and-seek" method, the reviewer is referring to erasure methods mentioned in Lines 96-98 of the main paper.
> Both GNDRP and hide-and-seek methods aim to highlight new regions,
> but they operate in a different manner.
> Hide-and-seek methods introduce additional convolution layers,
> forcing these layers to learn new information from a feature from which certain regions have been erased.
> In contrast, GNDRP forces the backbone network to learn new information of non-discriminative regions,
> without introducing new convolution layers; this makes GNDRP lighter than hide-and-seek methods.
>
>
> **Q3. Results in Figure 3.**
>
> A3. The results in Figure 3 were obtained by RIB-GNDRP.
> We will add a figure to visually compare localization maps obtained by RIB-GAP and RIB-GNDRP.
> We observed that localization maps obtained from CAM, RIB-GAP, and RIB-GNDRP showed gradual and noticeable improvement.
>
>
> **Q4. Generalization to Grad-CAM.**
>
> A4. Thank you for your suggestion. As mentioned in Section 3.1 of [ref1],
> CAM and Grad-CAM are identical for a CNN where global average pooled features are fed directly into the softmax layer.
> So, for ResNet, Grad-CAM is essentially the same as CAM.
> Therefore, we experimented with Grad-CAM++ [ref2], and our method improved it by 4.2 mIoU (47.7 &#8594; 51.9).
>
> [ref1] Selvaraju et al. "Grad-cam: Visual explanations from deep networks via gradient-based localization." IJCV, 2019.
>
> [ref2] Chattopadhay et al. "Grad-cam++: Generalized gradient-based visual explanations for deep convolutional networks." WACV, 2018.

---

> > ### Comment · Reviewer_cUpz · 2021-08-15
> > **Thanks for the response**
> >
> > I appreciate authors' responses to my comments. They have addressed most of my concerns.
> >
> > Hide-and-seek [a] is a kind of data augmentation method that does not require extra layers.
> >
> > [a] Hide-and-Seek: Forcing a Network to be Meticulous for Weakly-supervised Object and Action Localization, ICCV 2017
> >
> > I have also read comments from other reviewers and I keep my initial rating.

---

### Official Review · Reviewer_NvrK · 2021-07-16

**Rating:** 7
**Confidence:** 5

**Summary:**

This work addresses a limitation of CAMs (Class Activation Mappings) in the context of weakly supervised semantic segmentation (W3S) taking the information bottleneck perspective. The reason for the CAMs being highly selective, it argues, lies in the use of double-sided non-linearities (e.g. sigmoid) in the final layer of the classification network. The work removes this non-linearity and develops a strategy of obtaining high-recall attention maps through a one-shot adaptation process. Used as initial masks in a SotA W3S pipeline, these masks impressively increase the final segmentation accuracy achieving new state-of-the-art on established benchmarks.

**Limitations And Societal Impact:**

Limitations seems to be scattered throughout Sec. 4, which does not give a clear impression that could be useful for follow-up work. I recommend offering a discussion on the limitations in a more prominent way. The work does not discuss the potential societal impact which needs to be addressed in the revision.

**Main Review:**

*Originality.*
The approach proposed in this work has a number of novel and interesting elements. However, the problem with CAMs and the classification providing only “discriminative” regions is well-known and some attempts to address were not discussed or compared, such as LSE [A] and nGWP [B]. In particular, the motivation in [B] is not so dissimilar from the one in this work, as it also encourages the increased mask size through a loss term. Adding a discussion and a comparison to this alternative approach would make this work stronger.

[A] Pinheiro and Collobert. From image-level to pixel-level labeling with Convolutional Networks. CVPR 2015.
[B] Araslanov and Roth. Single-Stage Semantic Segmentation From Image Labels. CVPR 2020.

*Quality.*
The work is well-executed. I particularly appreciate the experiments both in the “labels-only” and “+saliency” setups on VOC-12, as well as experiments on MS-COCO. Model sensitivity to hyperparameters $m$, $\tau$, $K$ and $B$ are well-studied.

One remark however:
I do not see the necessity of introducing the circles and squares in the motivating experiment in Sec. 3.1. $G_6$ in Fig. 1 shows perfectly well that only a fraction of the digit is relevant for the final layer, whereas in Fig. 2 we now observe the tendency towards uncovering the full extent of the digit. So, why do we need the circles and squares? In natural images those may represent spurious correlations (e.g. an umbrella, which often co-occurs with the class “person”), hence the method will only pick up on those, as this toy experiment suggests.

*Clarity.*
The manuscript is well-written and follows a clear structure. The analysis and qualitative examples are insightful.

Minor issues are:
- ll. 134-135 “significant bottleneck”; “amount of information … will be limited”: these points is hard to agree with, since no measure is introduced. I’d suggest to make it relative to one-sided non-linearities, for example, to connect it with the discussion in Sec. 2.1.
- I’d consider renaming HGR, since it does not reflect the custom threshold of 0.3 (l. 164). As a result, Fig. 1 (c) and 2 (b) are unclear without a reference to the text.
- Please, add numbers to the equations on p. 5.

*Significance.*
The paper offers an interesting perspective on addressing the limitations of CAMs on which virtually all weakly supervised semantic segmentations  rely. The RIB process is relatively straightforward, the experiments demonstrate clear benefits. Future works on W3S and, perhaps, attribution methods, may find this technique useful.

**Post-rebuttal update**

I thank the authors for their response, and I keep my initial recommendation to accept this work. Nevertheless, I hope the design of the toy experiment can be still re-considered in the camera-ready version in view of my remark on spurious correlations.

**Time Spent Reviewing:**

3

---

> ### Author Response · Authors · 2021-08-10
> **Response to Reviewer NvrK**
>
>
> We appreciate the reviewer's insightful comments and suggestions on our manuscript. Below, we address the reviewer's main concerns.
>
>
> **Q1. Missing References**
>
> A1. Thank you for recommending papers [A] and [B].
> At first, GNDRP may appear similar to LSE [A] and nGWP [B] in that all three methods modify the pooling layer in the training time to obtain better localization maps.
> However, GNDRP operates in the opposite way.
> nGWP [B] aggregates features with weights obtained by a mask, which indicates the highly activated regions.
> Additionally, [B] increases the mask size through a regularization loss term.
> In contrast, GNDRP excludes highly activated regions, encouraging non-discriminative regions to be further activated.
> We will add more discussion and comparison to these methods in the revision.
>
>
> **Q2. Necessity of introducing circles and squares in the motivating experiments.**
>
> A2.
> We wanted to quantitatively measure the amounts of information conveyed by discriminative ($R_{D}$),
> non-discriminative but class-relevant ($R_{ND}$), and class irrelevant ($R_{BG}$) regions.
> However, if we only use digits, it is difficult to precisely distinguish between discriminative and non-discriminative areas within the region of a digit.
> Therefore, the introduction of circles and squares as $R_{ND}$ enables quantitative measurement of the information flow
> using HGR for each region (Figures 1(c) and 2(b)).
>
>
>
> **Q3. Spurious background correlations.**
>
> A3. As the reviewer correctly pointed out,
> spurious correlations between the target class and the background (e.g., boat-sea and train-rail) may exist in natural images.
> We would like to address this concern in cases of toy images and natural images separately.
>
> - (Toy images) To take a closer look at the spurious correlation, we design a new toy experiment. We define two different settings.
>   - Spurious correlation setting: We add a square to 70% of the images containing the digit '2', and also add a square to 20% of images containing the digit '8'.
> In this case, the square and the digit '2' frequently occur together,
> and thus they are highly correlated.
> However, this high correlation between the square and the digit '2' is spurious because the square also appears with the digit '8' occasionally.
> This is similar to an example of the spurious correlation between boat and sea: they frequently occur together,
> but the sea also occasionally appears with a bird.
>
>   - Class-relevant setting: We add a square to 70% of the images containing the digit '2', but we do NOT add a square to the images containing the digit '8'. In this case, since the square appears only with the digit '2', we can assume that the square is a part of the digit '2'.
>
>   We train classifiers under each setting and conduct the RIB process for each classifier.
> The table below shows HGR values of $G_6$ during the RIB process in each setting, similar to Figure 2(b).
> In the class-relevant setting, HGR values rapidly increase as the RIB process progresses, meaning that information flow in the square region is greatly increased.
> In contrast, in the spurious background setting, the increase in the HGR values is less prominent.
> It is thus reasonable to believe that our method can handle the spurious background effectively.
>
>
> |Setting|k=0|k=3|k=6|k=9|
> |----|:---:|:---:|:---:|:---:|
> |Class-relevant setting, HGR|0.17|0.35|0.55|0.64|
> |Spurious background setting, HGR|0.03|0.04|0.10|0.21|
>
>
>
> - (Natural images)
> Although RIB may activate some portion of the spurious background,
> the amount of discovered areas correctly belonging to the foreground is noticeably more significant.
> We support this argument empirically by measuring precision, recall, and F-score,
> which are reported in the table below.
> As the reviewer notes, our method produces a high-recall localization map but also improves precision.
>
>
> |Method|Precision|Recall|F-score|
> |----|:---:|:---:|:---:|
> |CAM |66.0|66.4|66.2|
> |Chang et al. (CVPR 20) [6] |61.0|77.2|68.1|
> |SEAM (CVPR 20) [51] |66.8|76.8|71.5|
> |RIB (Ours) |67.3|78.9|72.6|
>
>
> - To further analyze the effect of RIB on the spurious background, we present class-wise seed improvement by our method and other recent methods.
> We select two representative classes, boat and train, which are known to have spurious background correlation (boat-sea and train-rail).
> The table below shows that RIB can improve the localization quality (mIoU) even for classes known to have a spurious foreground-background correlation.
>
>
> |Method|Boat|Train|
> |----|:---:|:---:|
> |CAM |35.3|51.3 |
> |Chang et al. (CVPR 20) [6] |34.1|53.1|
> |SEAM (CVPR 20) [51] |31.0|54.2|
> |RIB (Ours) |38.8|55.1|
>
>
>
> **Q4. Minor issues**
>
> A4. Thank you for your detailed suggestions.
>
> - We will modify the sentence the reviewer mentioned into a comparative expression: for example,
> "Therefore, the final layer parameterized by w has a significant bottleneck" &#8594;
> "Therefore, the final layer optimized with a double-sided saturating non-linearity has a more severe bottleneck than that optimized with a single-sided one"
>
> - We will change HGR to HGR$^\gamma$, where $\gamma$ is the custom threshold, and modify Figures 1(c) and 2(b) accordingly.
>
> - We will add equation numbers to equations on page 5.

---

### Official Review · Reviewer_Az2H · 2021-07-18

**Rating:** 6
**Confidence:** 3

**Summary:**

This paper is about semantic image segmentation training with image-level labels. The method is using an auxiliary loss to help to find a larger localization area for pixel-wise training. Experiments are done on Pascal VOC2021 and MSCOCO.

**Ethical Concerns:**

No.

**Ethics Review Area:**

["I don’t know"]

**Limitations And Societal Impact:**

Yes.

**Main Review:**

1.A flowchart and an Algorithm to describe the whole method is absent.

2.This paper claims the information bottleneck or Data Processing Inequality is important for recognition and remove the last sigmoid. For a L-layer network with same resolution, ResNet-v2 style network has absolutely less information loss than ResNet-v1. However, we often find ResNet-v1 behaves better than ResNet-v2. It seems the information loss is not that important? Or what kind of suggestion could this concept provide to design a better network structure?

3.In page 5 Line 200, “Note that for each iteration, B−1 samples are randomly selected, while x is fixed.” This sentence is confusing.

4.In Table 4 it seems the backbone of proposed method is only ResNet-101. It is necessary to include more backbone, such as ResNet-50, Efficient, Transformer, etc., to prove the robustness of the method.


**Time Spent Reviewing:**

1

---

> ### Author Response · Authors · 2021-08-10
> **Response to Reviewer Az2H**
>
> We appreciate the reviewer's insightful comments and suggestions on our manuscript. Below, we address the reviewer's main concerns.
>
>
> **Q1. Flowchart and Algorithm.**
>
>
> A1. Thank you for your suggestion.
> We will add a flowchart and a pseudo algorithm to provide a more informative overview of the proposed method.
>
> Here, we present a simple pseudo algorithm.
>
> Requirements: Dataset $D$, classifier weights $\theta=[\theta_f, w\]$, feature extractor $f(;\theta_f)$, final classification layer's weight $w$
>
> 1: Train an initial classifier with $\mathcal{L}_{BCE}$, and its resulting weights are $\theta_0$
>
> 2: For each image $x \in D$:
>
> &nbsp;&nbsp;&nbsp;&nbsp;&nbsp;&nbsp;&nbsp;&nbsp; Define the map $\mathcal{M}$ of $x$
>
> &nbsp;&nbsp;&nbsp;&nbsp;&nbsp;&nbsp;&nbsp;&nbsp; For each RIB iteration $k$ ($1 \leq k \leq K$):
>
> &nbsp;&nbsp;&nbsp;&nbsp;&nbsp;&nbsp;&nbsp;&nbsp;&nbsp;&nbsp;&nbsp;&nbsp;&nbsp;&nbsp; Randomly sample $B-1$ images $x_1, ..., x_{B-1}$ from $D$ and construct a batch $X = [x, x_1, ..., x_{B-1}]$
>
> &nbsp;&nbsp;&nbsp;&nbsp;&nbsp;&nbsp;&nbsp;&nbsp;&nbsp;&nbsp;&nbsp;&nbsp;&nbsp;&nbsp; Compute $F = f(X)$ and $Y=w^T \text{GNDRP}(F)$
>
> &nbsp;&nbsp;&nbsp;&nbsp;&nbsp;&nbsp;&nbsp;&nbsp;&nbsp;&nbsp;&nbsp;&nbsp;&nbsp;&nbsp; Update weight: $\theta_k = \theta_{k-1} - \lambda \nabla_{\theta_{k-1}} \mathcal{L}_{RIB}(Y)$
>
> &nbsp;&nbsp;&nbsp;&nbsp;&nbsp;&nbsp;&nbsp;&nbsp;&nbsp;&nbsp;&nbsp;&nbsp;&nbsp;&nbsp; Compute CAM as in Eq. (2): $\mathcal{M}$ &#8592; $\mathcal{M} + w^T F$
>
> &nbsp;&nbsp;&nbsp;&nbsp;&nbsp;&nbsp;&nbsp;&nbsp; $\mathcal{M}$ &#8592; Normalize($\mathcal{M}$)
>
>
>
>
> **Q2. Questions on information loss.**
>
> A2. We assume that by "ResNet-v2," the reviewer is referring to pre-activation ResNet [ref1].
>
> -  We argue that a DNN with less information loss (large information flow) does not necessarily have to perform better than that with large information loss (restricted information flow).
> In fact, some previous works [1, 14, 50] argue that the information flow should be effectively restricted to achieve a better generalization ability.
> What is more important to the performance of a DNN than the overall amount of information is effective suppression of task-irrelevant information and preservation of task-relevant information.
> However, our goal is to improve the quality of localization maps, not to design a good classifier.
> As we have discussed in Sections 2 and 3, localization requires more information than classification,
> which motivated us to propose RIB.
>
> - The main focus of our work lies in increasing the amount of information flow within a single, fixed backbone through the proposed training technique.
> Therefore, measuring the difference in the amount of information flow across various backbones and
> choosing the optimal network structure for the target task deviate from our focus.
>
>
> [ref1] He et al. "Identity mappings in deep residual networks." ECCV, 2016.
>
>
> **Q3. Confusing sentence.**
>
> A3. We are sorry for the confusion.
> If the initial model is adapted only to $x$ during the RIB process, it can be easily over-fitted to $x$.
> Therefore, we utilize a separate set of randomly sampled $B-1$ images together with $x$ to avoid over-fitting.
> At each RIB iteration, we newly sample those $B-1$ images, while $x$ remains unchanged.
> For a clearer description of Line 200, please refer to the pseudo-algorithm in A1.
>
>
> **Q4. Performance with more backbones in Table 4.**
>
> A4. Following the reviewer's suggestion, we conduct experiments with ResNet-50 and obtain 42.4 mIoU.
> The results obtained using ResNet-50 offer a fairer comparison with CONTA [58], which also uses ResNet-50.
> Our method produces 9.0\%p better mIoU than that of CONTA.

---

> > ### Comment · Reviewer_Az2H · 2021-09-13
> > **Thanks for the response**
> >
> > I appreciate authors' responses to my comments. Some of my concerns have been addressed and I keep the initital rating.

---

### Author Response · Authors · 2021-08-10
**General Response**

We would like to thank the reviewers for their thoughtful comments on our submission.
In this author response, we address the questions raised by the reviewers
and indicate how we will revise our paper to reflect the offered comments.
We are glad that our paper is easy to follow (R2, R3, R4), demonstrates extensive experiments (R2, R3, R4),
provides an interesting and intuitive perspective on well-known limitations of CAMs (R2, R3, R4), proposes a novel solution (R2, R4), and shows a strong performance (R2, R3, R4).


As far as the limitations and the potential societal impact of our work are concerned,
we will make sure to include them in the revised manuscript.

- Limitations: We are sorry if the reviewers felt that we had not discussed the limitations of our method sufficiently.
In the revision, we will discuss our limitations including the presentation of failure cases.

- Societal impact: Object segmentation without the need for pixel-level annotation will save resources for research and commercial development.
It is particularly useful in fields such as medicine, where expert annotation is costly.
However, there are companies that provide annotations for images as a part of their services.
If the dependence of DNNs on labels is reduced by weakly supervised learning, these companies may need to change their business models.

---

### Decision · Program_Chairs · 2021-09-27

**Decision:**

Accept (Poster)

**Comment:**

The submission proposes an improved method for computing class activation maps, which are then used in weakly (image level label) supervised segmentation.  The main problem with class activation maps are that they often only highlight a small portion of an object.  Addressing this issue is the main focus of the paper, and does so with a different classification loss and a different global pooling layer.  The reviewers were unanimous in their opinion that the paper is above the bar for acceptance at NeurIPS, and appreciated the extensive experiments and utility of the setting.